# Memristor-based storage system with convolutional autoencoder-based image compression network

Yulin Feng [1,2], Yizhou Zhang[1], Zheng Zhou [1], Peng Huang [1 ✉], Lifeng Liu [1 ✉], Xiaoyan Liu[1] & Jinfeng Kang [1]

The exponential growth of various complex images is putting tremendous pressure on storage systems. Here, we propose a memristor-based storage system with an integrated near-storage in-memory computing-based convolutional autoencoder compression network to boost the energy efficiency and speed of the image compression/retrieval and improve the storage density. We adopt the 4-bit memristor arrays to experimentally demonstrate the functions of the system. We propose a step-by-step quantization aware training scheme and an equivalent transformation for transpose convolution to improve the system performance. The system exhibits a high (>33 dB) peak signal-to-noise ratio in the compression and decompression of the ImageNet and Kodak24 datasets. Benchmark comparison results show that the 4-bit memristor-based storage system could reduce the latency and energy consumption by over 20×/5.6× and 180×/91×, respectively, compared with the server-grade central processing unit-based/the graphics processing unit-based processing system, and improve the storage density by more than 3 times.

With the development of artificial intelligence and the Internet of Things (IoT), the amounts of data generated from data centers, automotive, and edge devices have increased dramatically. As a result, higher requirements for data storage have been put forward, including bit density/chip and memory capacity[1–3]. The emergence of 3D NAND has resolved the density limitations of conventional planar NAND through 3D integration techniques, with improved scaling and bit density[4]. However, the further improvement of density limitations will ultimately be constrained by the storage architecture, devices, and process-induced reliabilities[5,6]. Therefore, it is necessary to explore new attractive storage strategies. Storage systems that compress data are being keenly sought after to alleviate the storage pressure caused by large amounts of data[7], especially for applications that can tolerate partial loss after data reconstruction. Image compression is a main branch of data compression that focuses on digital images, encoding the original image with fewer bits to reduce the storage and transmission costs[8,9]. Traditional lossy image compression techniques such as joint photographic experts group (JPEG) and JPEG2000 usually include domain transformation, quantization, and entropy coding modules[10–12].

Recently, a deep learning-based image compression technique has emerged as an alternative. It has been widely studied in image compression due to its superior performance and ability to learn practical information from different images in various scenarios[13–17]. Moreover, the performance of the deep image compression network is comparable with, or even better than, that of JPEG, JPEG2000, and better portable graphics (BPG)[14,15,17,18]. Integrating such a reconfigurable compression network, consisting of an encoder and decoder, into the storage system is an innovative and effective way to improve system efficiency and image storage density for edge devices and cloud servers.

For the hardware implementation of deep neural networks (DNNs), the memristor crossbar-based processing-in-memory architecture has proven to be an efficient approach due to its high

[1]School of Integrated Circuits, Peking University, 100871 Beijing, China. [2]Key Laboratory of the Ministry of Education for Optoelectronic Measurement Technology and Instrument, Beijing Information Science & Technology University, 100192 Beijing, China. ✉e-mail: phwang@pku.edu.cn; lfliu@pku.edu.cn

parallelism and energy efficiency[19–24]. A vital advantage of this integration structure is that it cannot only store weights and realize vector-matrix multiplication (VMM) in one step to accelerate computations of neural networks[25–30] but also can be used as a general-purpose non-volatile memory for high-density data storage[31–33]. Utilizing memristors for computing and storage in the same physical unit can fully leverage this advantage. Recent studies have reported using memristor arrays to experimentally demonstrate JPEG-based image compression[34,35] and compressed data storage[36,37]. However, no study has been conducted on fully memristor-based storage in which the data compression, decompression, and storage are fully hardware implemented based on the memristor arrays.

This study proposes a memristor-based storage system integrated with the near-storage in-memory computing for data compression/decompression and multi-bit storage banks. We use the 4-bit memristor to experimentally demonstrate the function of the proposed system, including encoding, data storage and retrieval, and decoding. We design a convolutional autoencoder (CAE) network as a codec for natural image compression and an equivalent transformation for transpose convolution to accelerate the decoding process. We propose a step-by-step quantization aware training scheme for device-algorithm codesign to improve the quality of the reconstructed images. The system can achieve a high (>33 dB) peak signal-to-noise ratio (PSNR) in the compression and decompression of the ImageNet and Kodak24 datasets. The Top5 accuracy for recognizing the reconstructed ImageNet dataset with ResNet34 only decreases by 0.06% compared with the original dataset, indicating the high fidelity of the reconstructed images. The evaluation results indicate that the proposed system effectively reduces latency and energy consumption by more than 20× and 180× compared with the server-grade central processing unit (CPU)-based processing system, 5.6× and 91× compared with the graphics processing unit (GPU)-based processing system, and 5.6× and 30× compared with the application-specific integrated circuit (ASIC)-based and non-near-storage in-memory processing systems. In addition, the storage density could improve by more than 3 times. This study paves an efficient path for developing an efficient and high-density storage system tailored for the image data.

## Results

### Memristor-based storage system with image compression network

Figure 1 schematically illustrates the proposed storage system with the near-storage in-memory computing architecture and its implementation based on memristor arrays. The system contains multiple storage banks and in-memory computing banks to compress and decompress the image data (Fig. 1a). The computing and storage cores are distributed across different banks for the computation of the codec network and the storage of the compressed data, respectively (see Supplementary Fig. 1). The images to be stored are firstly compressed by the encoder of the codec, which is realized by using convolution operations to reduce the dimensionality of the input image while extracting its main spatial features. As a result, the compressed representations are obtained and can be stored in multi-bit memristor arrays. When the image needs to be retrieved, the stored data is read from the storage arrays and decompressed by the decoder, which utilizes multi-step transpose convolution operations to restore the original image at each pixel. During the reconstruction process, the size of the image and the pixel information on the corresponding image position can be restored. Considering the memristor's the multi-level characteristics and the array's crossbar integration structure, using the memristor arrays to implement the proposed storage system is very advantageous and attractive. The image compression network with different types and scales can be configured into the system according to the specific application scenarios and available hardware overhead. Thanks to the adaptability and robustness of neural networks, the system integrated with the deep learning-based compression network is suitable for compressing and decompressing diverse and complex images.

To demonstrate the functionality of the storage system, we constructed a two-layer CAE network (Fig. 1b) within the system. The network contains one convolution layer in the encoder and one transpose convolution layer in the decoder, responsible for the input image compression and the stored data decompression, respectively (see Methods). The inputted high-resolution test image needs to be segmented into a series of sub-images with the same size as the training set, which is then sequentially input into the network. Figure 1c and d schematically show the typical convolution and transpose convolution and the corresponding events in the memristor array. The blue square in the figure represents an input case that slides over the input to achieve a reduction or expansion of the image in dimensionality. We adopted the bit-slicing encoding technique for multi-bit input of the memristor array (see Supplementary Note I). This technique first involves converting the input into a sequence of binary voltage pulses, which are then flattened before inputting into the word lines (WLs) of the memristor array by using time division. A signed kernel weight is mapped to a pair of memristors with differential conductance for weight transfer. Each memristor pair represents a different weight bit and has a different weight coefficient ($k^n$), where k is related to the available conductance levels of the memristor, and n refers to the bit number. The output Y can be obtained by multiplying the source line (SL) output and weight coefficient.

Different from the neural network used for image recognition, which determines the classification of input by comparing the probability of the network outputs, the main target of image compression is to restore the specific value at each spatial pixel. Therefore, higher weight quantization accuracy is required for better-quality reconstructed images. Each weight is first converted into a series of bit numbers and then mapped to multiple pairs of memristors to achieve the mapping of the high-precision weights (see Supplementary Fig. 2). After encoding the pulses and inputting them into the WLs, the currents of the two different SLs are sensed and subtracted as the output of corresponding weight bit. The final result is obtained by summing the weighted outputs of different pairs of SLs.

### Implementation of storage system with memristor arrays

For hardware implementation of the storage system, multi-bit memristor arrays were fabricated to perform the computation of the encoder/decoder networks and storage of the compressed data. The testing set-up with packaged 1-transistor 1-memristor (1T1M) chip and the photographs of the die and array structure are shown in Fig. 2a. A 4 kb (256×16) crossbar array with TiN/TaO$_x$/HfO$_x$/TiN structure is designed and fabricated on top of the CMOS circuits (see Methods)[38,39]. The array design of this size is conducive to reducing the complexity and area overhead of peripheral circuitry (see Supplementary Fig. 3 and Note II). The memristors in the same column share the same bit line (BL) as top electrode and the same SL connecting to the source of transistors, while those in the same row possess a common WL connecting to the gate of transistors. The transistors provide efficient control over multi-level conductance tuning with sufficient accuracy. Programming and computing are performed on a custom-built testing system connected to the test board, and all signals are generated off-chip (see Methods). Excellent inter- and intra-cell cycling uniformity can be achieved, as shown in Fig. 2b and Supplementary Fig. 4. By using the bidirectional incremental gate voltage in combination with fixed SET and RESET pulses (see Methods and Supplementary Fig. 5)[39], the memristor can be precisely programmed to an arbitrary conductance state within a predefined error range. Figure 2c shows the distribution of 1600 memristor cells in 16 conductance states, where no overlap is observed between neighboring curves. To verify the feasibility and validity of the array operation and weight

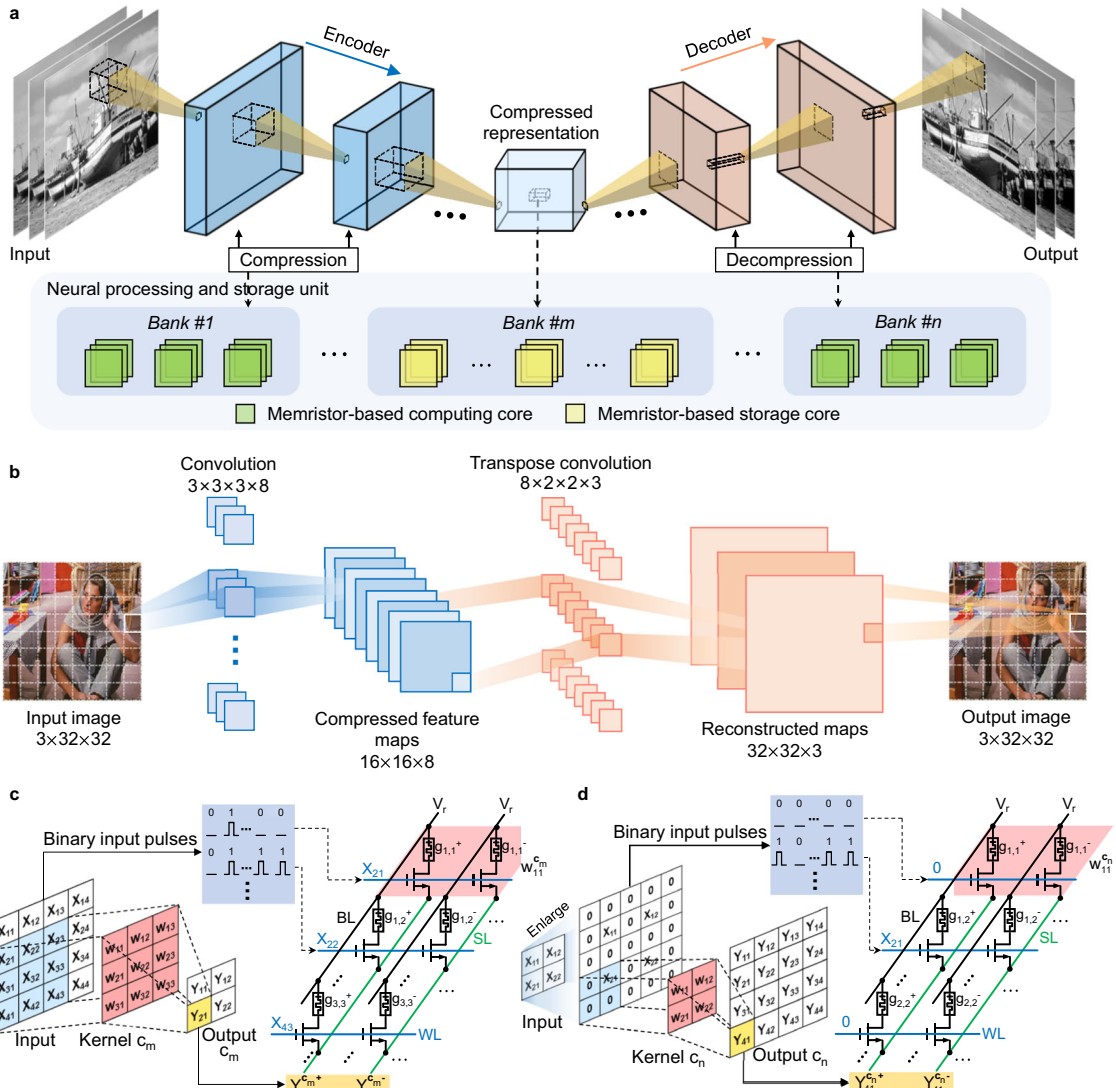

**Fig. 1 | Memristor-based high-density storage system. a** Schematic illustration of the proposed near-storage in-memory processing system implemented with memristor-based cores. The upper part shows a multi-layer image compression network. The compression and decompression processes are generally structurally symmetric to ensure that the resolution of the reconstructed image is consistent with that of the input. Once the network completes training, it can be used for any input to acquire a compressed representation. The lower part shows the implementation of the system with memristor-based cores. The green and yellow squares represent the computing and storage cores, respectively. The computing banks and storage banks are located in the neural processing and storage unit. **b** A constructed two-layer convolutional autoencoder (CAE) network that contains eight kernels in convolution layer for compression and three kernels in transpose convolution layer for decompression. Implementation of (**c**), convolution and (**d**), transpose convolution with memristor arrays. The blue, red, and yellow rectangles represent the selected elements of the input, kernel weight, and output, respectively. Through the time-division input of the image and multiply-accumulate operation of the source line (SL) output, the pixel value of the output feature map can be obtained. During the entire computing process, a constant read voltage ($V_r$) is applied to all bit lines (BLs), and the SLs are grounded. Kernel $c_m$ and $c_n$ represent the $m_{th}$ convolution kernel and $n_{th}$ transpose convolution kernel, respectively. Output $c_m$ and $c_n$ correspond to the feature maps output by the $m_{th}$ convolution kernel and $n_{th}$ transpose convolution kernel, respectively.

mapping schemes for VMMs, a typical convolution experiment with Sobel operators for edge extraction is carried out in the 1T1M array (see Supplementary Fig. 6). The high precision of the computation results in correlation plots of ideal VMM and experimental VMM indicates the effectiveness of the schemes.

Figure 3a illustrates the overall operation flow of the system implementation with memristors. The data retrieval, transmission, and reading can be realized through the interactive control of 1T1M chips, microcontroller (MCU), and software. Once the system is trained, the weights are quantized and mapped to the memristor arrays. The convolution and transpose convolution are performed by the VMM of the memristor array, which enables high computation efficiency. For the process of compressed data storage, the analogue output of the encoder network is first quantized based on the limited available conductance states of the memristor and then written into the storage array. The write-verify scheme is applied for weight mapping and data writing.

To implement the function of the system with high performance, we propose effective strategies for algorithm-device codesign for decompression processes and introduce the digital readout scheme[40] into the CAE network. The PSNR value, a typical metric for image compression[15,41], is used to evaluate the quality of the images after decompression (see Methods). For the weight mapping process, a convenient and universal method is to directly quantize the weights to different levels to match the multi-bit memristor. In this manner, the performance of the CAE network is severely affected by the quantization accuracy of the weight or the compressed data (see Supplementary Fig. 7). The design space exploration is performed with

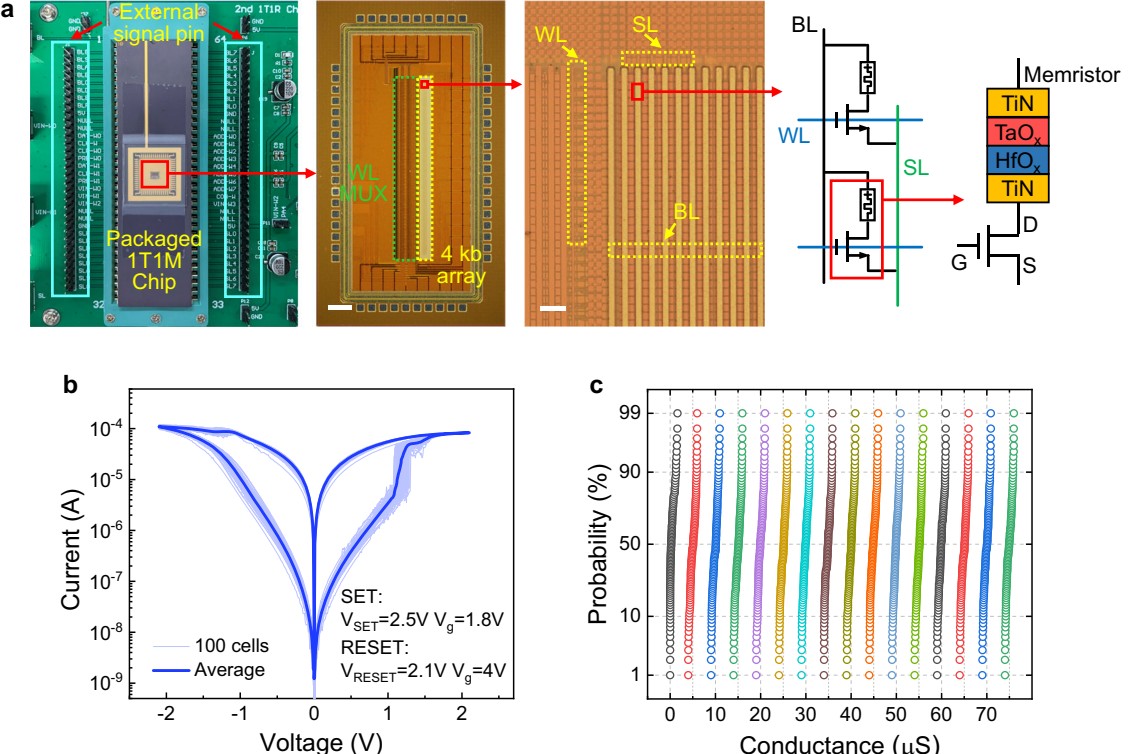

**Fig. 2 | Memristor array and electrical performance. a** Testing set-up with packaged 1-transistor 1-memristor (1T1M) chip, optical microscopic image of the memristor array, and the integration and cell structure of the memristor. Word line multiplexer (WL MUX) is integrated inside the chip, while the control components for the bit line (BL) and source line (SL) are integrated on the test board and connected to the chip via bonding pads. G, D, and S represent the gate, drain, and source of the transistor, respectively. The scale bar in the second and third images from the left is 200 μm and 10 μm, respectively. **b** Statistics of the quasi-static direct current (DC) cycling behavior of 100 memristor cells. **c** Cumulative probability distribution of 1600 memristor cells with respect to 16 independent conductance states. Read voltage ($V_r$) is set to 0.2 V during programming.

different quantization precisions and CAE network structures for choosing applicable precisions of weight and compressed data (see Supplementary Fig. 8). Considering the hardware overhead and the quality of the reconstructed image, we chose 8-bit and 6-bit precision to quantify the weights and data after compression, respectively. To alleviate the impact of the deviations caused by the direct quantization, the CAE network is retrained with quantized weights and compressed data. The quantized weights and activations are used in the forward propagation, while 32-bit floating-point numbers are utilized in the back-propagation process to update the weight[23,42–44]. However, the performance improvement after retraining is limited, which might be mainly due to the direction of image dimension change being opposite during the encoding and decoding processes in the CAE network. Therefore, we propose a step-by-step quantization aware training scheme (Fig. 3b) in which the compressed data, encoder, and decoder are quantized and trained in sequence to adapt to the changes of the image size in different directions during encoding and decoding (see Methods). Based on the proposed quantization scheme, the performance of the system is significantly improved and approaches that of software as the number of network layers increases (Fig. 3c), which effectively suppresses the degradation of system performance caused by the quantization-induced deviations. It is worth noting that the PSNR value decreases with more codec layers, mainly due to partial information loss during downsampling operations in convolution computing process. This issue can be alleviated by optimizing the network structure or training scheme, such as nonlinear transformation[15] or content-weighted scheme[13] so that a large-scale network can be used to obtain a high compression ratio while achieving an accepted PSNR value.

When demonstrating image compression based on non-volatile memories (NVMs), the stored data is usually read out in analogue form and used for image reconstruction[35,36,45]. The readout conductance is then converted into voltage pulses with different amplitudes or widths and directly used for image decompression, which is discussed in detail in Supplementary Note III and Fig. 9. For memristors, the read-out conductance values from the storage array are distributed around the target conductance level (Fig. 3d). This will lead to deviations in the converted voltage pulses that are used in subsequent decoding computations, resulting in the deterioration of the quality of the reconstructed image. To eliminate this deviation, we utilized a digital readout method, which appears common for multi-bit memory systems, for hardware demonstration of deep learning-based image compression technique to improve the quality of reconstructed images. When using this scheme, the readout conductance value is first quantified and then input into the transpose convolution network bitwise from the most significant bit (MSB) to the least significant bit (LSB) to perform decompression computations. In this case, the distributed conductance values can be accurately quantified to the discrete target conductance level without needing other peripheral circuitry for data conversion operation. A typical storage and readout process are shown in Supplementary Fig. 10. It can be observed that the reconstructed pixel matrix is completely consistent with the quantized pixel matrix by utilizing the proposed method to read the compressed data, which is critical for high-quality image reconstruction. Figure 3e shows the statistical distribution of the error between the readout conductance and the target conductance using both readout methods. The digital readout can tolerate variations in the intermediate storage and reading process, thus preventing the

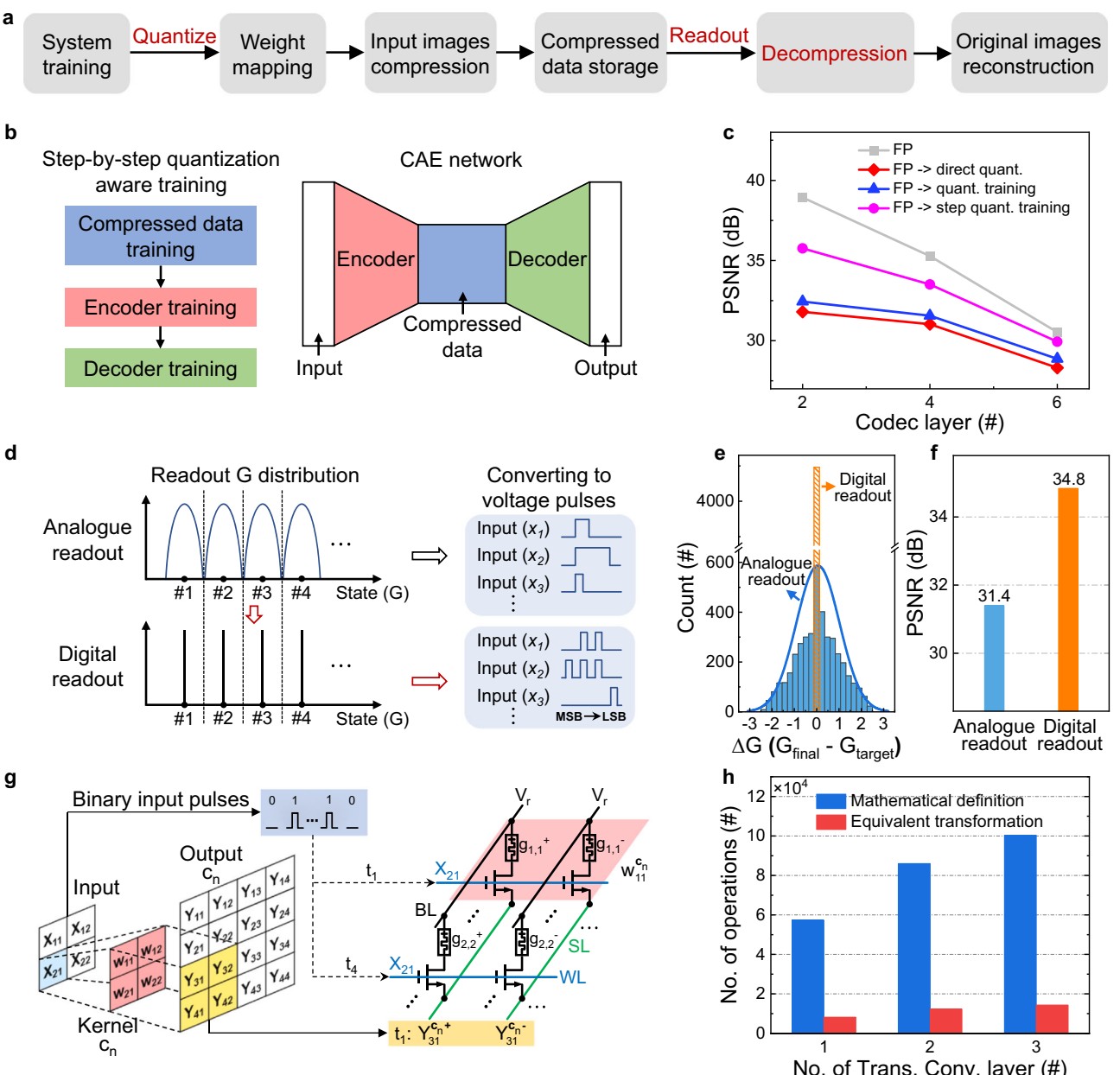

**Fig. 3 | System processes and performance optimization strategies. a** Operation flow of the proposed storage system. Fonts in red indicate optimized processes. **b** Sketch of the convolutional autoencoder (CAE) network and the corresponding training sequence of each part in CAE during step-by-step quantization aware training. **c** The quality of reconstructed images with respect to different quantization schemes. The proposed quantized training scheme demonstrates more significant improvement in network performance for a codec with fewer layers. FP and quant. refer to the floating point and quantization, respectively. **d** Schematic diagram of the analogue and digital readout and conversion processes. The conductance values by using digital readout are quantized through analog-to-digital converter (ADC), which is essentially in the form of binary and can be directly input to the decoder network. MSB and LSB refer to the most significant bit and the least significant bit, respectively. **e** Histogram of the error distributions obtained by

utilizing the analogue and digital readout methods. $G_{final}$ represents the measured conductance using either the analogue or digital readout method, while $G_{target}$ represents the target conductance. **f** Comparison of the peak signal-to-noise ratio (PSNR) values of the reconstructed images using two readout methods. **g** Schematic of the equivalent transformation of transpose convolution and its implementation in memristor array. The blue, red, and yellow rectangles represent the selected elements of the input, kernel weight, and output, respectively. **h** Number of operations with respect to different transpose convolution layers based on both mathematical definition and equivalent transformation. The transpose convolution with 1, 2, and 3 layers correspond to codecs with two, four, and six layers, respectively. Trans. Conv. is used as an abbreviation for transpose convolution in the figure.

introduction of errors in the readout conductance array. On the other hand, the error of the readout conductance in analogue reading conforms to the normal distribution due to the drift of the memristor conductance and the disturbance of the read circuit, which directly leads to the degradation of the quality of the reconstructed image, as shown in Fig. 3f. Intuitive results by using two readout methods can be

seen from the visualization of the reconstructed images (see Supplementary Fig. 11).

To achieve upsampling during decompression, the transpose convolution is carried out. In this process, the input first needs to add zero spacings between each element in the spatial dimension according to the mathematical definition, followed by step-by-step

convolution. However, this process leads to a lot of zeros being inserted into the input, requiring large amounts of multiplication and addition operations on these pixels during transpose convolution. As a result, a significant number of invalid operations are produced, increasing the computational load. To improve the efficiency of the decompression process, we designed an equivalent transformation derived from the definition of transpose convolution (see Supplementary Note IV), which directly utilizes the kernel to weight each element of the input (Fig. 3g), effectively avoiding invalid calculations caused by zero inputs and substantially reduces the number of operations required when hardware implemented by memristor array. As shown in Fig. 3h, after undergoing the equivalent transformation, the number of operations required for transpose convolution is reduced by a factor of 7 as compared to the mathematical definition.

For system training, the Cifar-10 dataset is used to train the CAE network before being applied to compress the tested natural images. Once the ex-situ trained convolution and transpose convolution kernel weights are transferred into memristor arrays, the functions of data compression, storage, and reconstruction can be demonstrated in 1T1M array. Figure 4 shows the step-by-step experimental results of the system, including weight mapping, image compression, reconstruction, and data storage and readout. The quantized kernel weights are accurately transferred to the corresponding memristor with small errors between the actual and target conductance. Benefiting from the excellent retention property of the conductance state, the conductance value after mapping has little fluctuation. The segmented input image is convolved by kernels, and eight grayscale images containing different feature information are achieved by array computation. After convolution, the analogue compressed feature map is quantized to 6 bits to acquire pixel

quantization matrix. In this case, each pixel information can be stored on two pairs of 4-bit memristor cells, which leads to a 3× improvement in storage density. Each compressed feature map can be stored in the storage array as compressed data and read out when needed for application. During reconstruction, the stored data are read out followed by computation with transpose convolution kernels to output three feature maps that correspond to the original RGB images. Since the output of the VMM is the accumulation of currents through a column, the small random drift of the cell conductance has little impact on it[46], which is critical for the image compression network that needs high accuracy of the reconstructed pixel information. Since the input is converted into bits from MSB to LSB by bit-slicing encoding, it should be noted that a large conductance drift in the memristor corresponding to MSB of the weight can still cause a large output deviation. Therefore, we use the programming scheme proposed in our previous work to maintain the conductance stability of the MSB weight[39].

As observed in the implementation results, there are no discernible differences between the reconstructed image in the experiment and that in the simulation, and both are consistent with the input image in terms of visualization degree, which includes object position, edge contour, and contrast. Although there is a slight degradation in the PSNR value (<1 dB) in experiment, which is probably arises from the quantization errors and deviations in the computing operations. But this degradation is still within an acceptable range for image compression. The successful demonstration of the storage system and the consistent results of the obtained feature maps between experimental and simulation manifest the feasibility of the proposed storage system, as well as the reliability of implementing deep learning-based image compression with multi-bit memristor arrays.

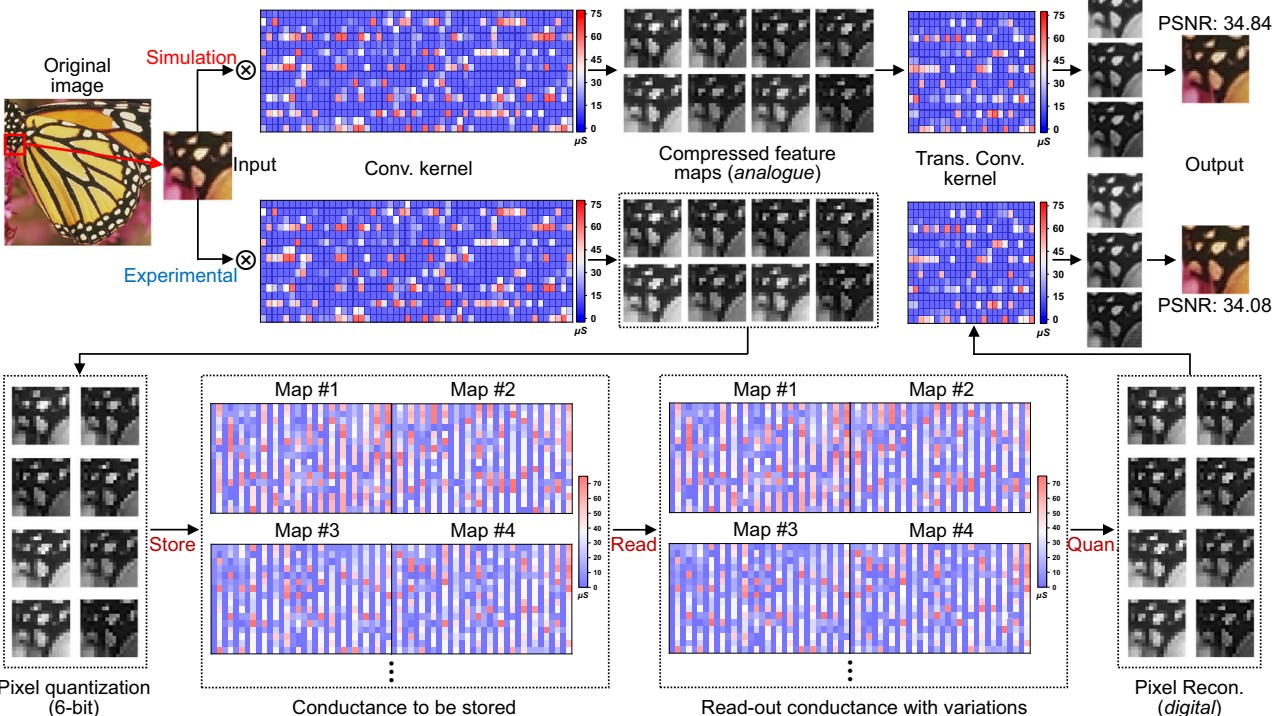

**Fig. 4 | Implementation of the memristor-based near-storage in-memory processing system.** Eight compressed feature maps are achieved with analogue values by convolving the input with convolution (Conv.) kernels. The feature maps are then quantized and stored into the storage arrays. The original image can be restored by performing a transpose convolution (Trans. Conv.) operation with the compressed feature maps (digital). The number of memristors required to map the weights of the transpose convolution kernel is much smaller than that used for convolution kernels, which means that feature extraction in CAE network is more critical. The resolution and pixel depth of the compressed 8 feature maps are 16 × 16 and 6-bit, respectively. Since the original image is 32 × 32 with 8-bit for 3 color channels, as a result, a compression ratio of 2 is achieved with the demonstrated system (see Methods). Quan. and Recon. are used as abbreviations for quantization and reconstruction, respectively. PSNR refers to peak signal-to-noise ratio.

## Performance evaluation of the storage system

The practicability and fidelity of the image data passing through the proposed near-storage in-memory processing system are vital metrics for data storage and subsequent practical applications. We deployed the ImageNet dataset and Kodak24 dataset, which contain richer images of daily life and various image types unrelated to the training dataset (Cifar-10), to examine the restoration degree of the trained storage system in the face of unseen images. Visualization results of the randomly selected sample images after compression are the same as those before compression (see Supplementary Figs. 12, 13), and all the images obtain >33 dB PSNR values, which proves the universality of the proposed storage system and its 'data class'-independent characteristic. Moreover, we used reconstructed and original ImageNet datasets to perform image recognition tasks with the ResNet34 neural network (see "Methods"). The ResNet34 neural networks are trained and tested with the original and reconstructed ImageNet datasets, respectively. The recognition results, shown in Fig. 5a, indicate that the network with the reconstructed dataset achieved almost the same Top5 recognition accuracy (91.36%) as the original dataset (91.42%) without performance degradation. We also examined the network performance when using different types of ImageNet datasets for training and test (see Supplementary Fig. 14). The accuracy slightly decreases (<1%) when the test set is different from the training set, which is probable due to the partial mismatch of the pixel values between two datasets and has little impact on practical applications.

We chose the VisualQA v2.0 dataset[47] to be stored to estimate the latency, energy consumption, and storage density of the proposed system. As shown in Fig. 5b, the required area for storing the dataset is greatly reduced after compression with the increase in the number of network layers and multi-bit capabilities of the memristor. To more fairly discuss the area overhead, the areas, including encoding/decoding, storage arrays, and peripheral circuitry used for compression/retrieval, are all considered during evaluation (see Methods). In this case, the storage density improves by a factor of 3 when using the constructed system in our demonstration (two-layer CAE network with 4-bit memristor), and a maximum storage density improvement of 18 times can be achieved if a six-layer CAE network and 8-bit memristor arrays[48] are adopted. It can be inferred from these results that by using deep neural networks with higher compression ratios or memristor arrays with more states, the storage density can be significantly improved (see Supplementary Fig. 15). However, the ensuing challenges should also be given due consideration. On one hand, as the degree of compression increases, the distortion of the reconstructed image will become more significant. On the other hand, the reliability of the multi-bit memristor will deteriorate with the increase in the number of conductance states. Therefore, the trade-off between storage density improvement and image quality should be carefully considered according to different application scenarios. Factors including algorithm, network structure, device characteristics, and selection of training dataset need to be co-optimized to minimize the loss of the reconstructed image while achieving a high compression ratio. Although integrating a deep image compression network into the storage system will cause some hardware overhead, it is insignificant compared with the hardware consumption for data storage.

The impact of the device variations on network performance is also assessed by introducing the normalized standard deviation (σ) to the memristor conductance (see Methods). The PSNR values of the different networks decrease as σ increases (Fig. 5c). This can be attributed to conductance deviations, which cause the pixel values of the reconstructed image to deviate from the corresponding values in the original image. We also discovered that the quality of the reconstructed image in the memristor-based storage system is more susceptible to weight conductance fluctuations than compressed data

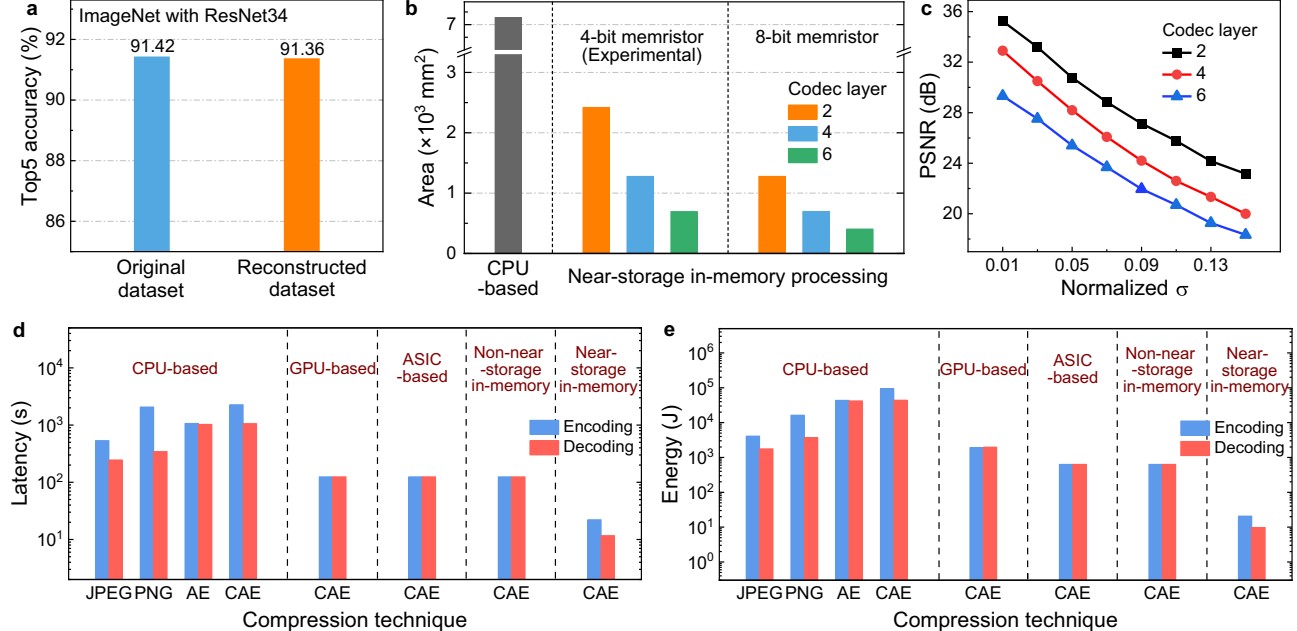

**Fig. 5 | Performance evaluation of the storage system. a** Top5 recognition accuracy of ResNet34 neural network that is trained and tested with the original and reconstructed ImageNet datasets. **b** Area comparison of server-grade central processing unit (CPU)-based processing system and proposed near-storage in-memory processing system for storing VisualQA v2.0 dataset. Different codec layers and memristor storage capacities are used in proposed storage system. The area of peripheral circuitry used for encoding/decoding and storage are taken into consideration. **c** The PSNR values of the reconstructed images with different codec layers as a function of normalized conductance standard deviations. **d** Latency and (**e**), energy comparison of server-grade CPU-based, graphics processing unit (GPU)-based, application-specific integrated circuit (ASIC)-based, non-near-storage in-memory, and near-storage in-memory processing systems for implementing compression techniques including joint photographic experts group (JPEG), portable network graphic (PNG), autoencoder (AE), and convolutional autoencoder (CAE).

(see Supplementary Fig. 16), which can be used as a guidance for system design.

We compared the latency and energy consumption of the proposed system with that of other competitive technologies, including JPEG, portable network graphic (PNG), autoencoder (AE), and CAE implemented in the server-grade CPU-based processing system, and CAE implemented in the GPU-based, ASIC-based, and non-near-storage in-memory processing systems (see Methods and Supplementary Fig. 17). Since traditional compression techniques usually contain various domain transformations and complex lossless coding schemes (such as Huffman coding), in most cases, the compression is better suited for execution on the CPU. As illustrated in Fig. 5d, e, the near-storage in-memory processing system effectively reduces the latency and energy consumption by more than 20× and 180×, respectively, compared with server-grade CPU-based processing system, 5.6× and 91×, respectively, compared with GPU-based processing system, and 5.6× and 30×, respectively, compared with both ASIC-based and non-near-storage in-memory processing systems. The significant improvements in energy efficiency can be mainly attributed to the integration of memristor-based computing banks and storages banks within the same unit, which effectively reduces the latency and energy consumption caused by circuit module interfaces and internal transfer during data movement. It should be noted that the latency and energy reductions of ASIC-based processing system is almost the same as those of non-near-storage in-memory processing system, which is due to the fact that in both systems, the overhead of data access during compression and decompression processes is much greater than that of data processing. The overall latency and energy consumption are dominated by the storage unit. Moreover, the impact of memristor multi-bit capability on boosting system performance is further discussed (see Supplementary Note V). The results illustrate that memristor devices with more storage capacities could effectively promote the overall efficiency of the system (see Supplementary Figs. 18, 19).

In Table 1, we summarized and compared the reported implementations of emerging NVM-based image compression techniques and storage after compression. Previous studies focused mainly on using memristors to only implement the frequency domain transformation step in traditional compression algorithms[34,35], or using phase change memory (PCM)/memristors only for storing compressed data[36,37,45]. Meanwhile, the restored image quality after decompression is not very high. Here, we experimentally demonstrate the fully memristor-based storage system equipped with image compression/decompression networks. Compression (convolution computing), compressed data storage and access, and decompression (transpose convolution computing) are all hardware demonstrated accompanied by achieving the high PSNR value (34.08 dB) of reconstructed images. The achievements provide an efficient avenue and serve as a guidance for developing efficient and high-density storage technology.

## Discussion

In this study, we proposed a memristor-based storage system that integrates a deep image compression network for image data storage, which is implemented based on a near-storage in-memory processing architecture with high efficiency and storage density. A two-layer CAE network was constructed in the storage system. All the system functionalities, including compression, decompression, and data storage and access, were hardware demonstrated with 4-bit memristor arrays and achieved a comparable PSNR (34.08 dB) to that in simulation (34.84 dB). We proposed a step-by-step quantization aware training scheme that improves PSNR value by 5 dB compared to the direct quantization scheme. In addition, we designed an equivalent transformation for transpose convolution, which dramatically reduces the amount of computation by more than 7 times. The benchmark comparison results illustrate that the memristor-based near-storage in-memory processing system can achieve reductions in latency and energy consumption of over 20× and 180×, respectively, when compared with the server-grade CPU-based processing system. Similarly, it achieves reductions of 5.6× and 91×, respectively, when compared with the GPU-based processing system, and 5.6× and 30×, respectively, when compared with the ASIC-based and non-near-storage in-memory processing systems. Considering the areas of encoding/decoding array, storage array, and peripheral circuitry, the storage density is increased by a factor of 3. It can be further improved to 18 times when employing a six-layer CAE network with 8-bit memristors. It is essential to carefully consider the trade-off between storage density and reconstructed image quality when designing the storage system. The trained storage system (with Cifar-10) is applied to compress and store the ImageNet and Kodak24 datasets. The high PSNR values (>33 dB) and clear visualization results exhibit the universality of the trained system and independent characteristics of the class of the data that is being stored, which is critical for a storage system. Image recognition tasks were performed in the ResNet34 neural network with the reconstructed ImageNet dataset, achieving a 91.36% Top5 accuracy which is comparable to that (91.42%) with the original dataset. The implementation of the memristor-based near-storage in-memory processing system provides an effective alternative for efficient and high-density storage of growing diverse image information, which can be further extended to the compressed storage of any data by using emerging and suitable deep learning-based compression networks with larger and more reliable memristor arrays.

## Methods

### Fabrication of 1-transistor 1-memristor array

The memristor is composed of $TiN/TaO_x/HfO_x/TiN$ structure. Standard CMOS foundry processes under 130 nm technology node are used to fabricate the transistors, metal interconnections, and vias. The remaining memristor stacks and top metal interconnections are completed in the laboratory. The memristor cell is formed on the drain of the transistor. An 8-nm-thick $HfO_x$ switching layer is deposited on the TiN bottom electrode using atomic layer deposition with water and tetrakis(dimethylamido)hafnium as precursors at 250 °C. A 45 nm-thick $TaO_x$ capping layer is then deposited by sputtering the Ta target in $Ar/O_2$ atmosphere, which acts as thermally enhanced layer to modulate the electric field and heat in $HfO_x$. The top electrode of TiN/Al are deposited using sputtering and electron beam evaporation. Finally, contact pads were formed through dry etching.

**Table 1 | Summary of hardware implementation of image compression based on emerging NVMs**

| Image compression technique | Compression | Decompression | Storage | Reconstructed image quality |
|---|---|---|---|---|
| JPEG[35] | Memristor (for DCT) | Software | – | – |
| _[34] | Memristor (for DFT) | Memristor (for IDFT) | – | 26.1 dB |
| Fully-connected AE[45] | Software | Software | PCM | 25.1 dB |
| Fully-connected AE[36] | Software | Software | Memristor | 30.3 dB |
| Convolutional AE (this work) | Memristor | Memristor | Memristor | 34.08 dB |

*DCT* discrete cosine transformation, *DFT* discrete fourier transformation, *IDFT* inverse discrete fourier transformation, *AE* autoencoder, *PCM* phase change memory.

## Experimental set-up and write-verify programming scheme

A customized measurement system has been built to read and write the 1T1M chip. The 1T1M chip is wire-bonded into a dual in-line package and mounted on a printed circuit board (PCB). Power signals and SL/BL gating chips are integrated on PCB, as only WL control circuits are fabricated into the chip. MATLAB scripts are written to control the measurements by communicating the MCU and external signal sources. For electrical characterizations, set-ups including Agilent B1500A, 81160 A, and 34980 A are utilized.

To program multi-bit memristors efficiently and reliably, we utilize the write-verify programming scheme based on bidirectional incremental gate voltage, in combination with fixed SET and RESET pulses ($V_{set}$=2.8 V, $V_{reset}$ = 3 V, $P_w$ = 100 ns). The read voltage is set to 0.2 V throughout the programming process. We set the defined error margin to ±1 µS and maximum programming pulse number to 100. During programming, the gradually incremental voltage is input to the gate of the transistor to apply a compliance current for conductance modulation. Once the measured conductance falls within the target conductance range, the programming successfully ends. Otherwise, the conductance modulation continues until the maximum pulse number is reached. We program 16 individual target conductance states onto randomly selected memristors, and each conductance state is programmed onto 100 cells. The target states are distributed within the switching window from <1 µS to 75 µS with a uniform interval of 5 µS. During the experiment, the conductance of the memristor array is programmed from 0 µS up to 75 µS.

## Convolutional autoencoder network

The CAE network is an end-to-end unsupervised learning model that learns optimal convolution and transpose convolution operators to minimize the reconstruction error. The CAE network can be considered as two transformations: an analysis transformation with an encoder function $y = f_\alpha(z)$ and a synthesis transformation with a decoder function $\hat{z} = g_\beta(y)$. The $z$, $y$, and $\hat{z}$ represent original images, latent representation, and reconstructed images, respectively. $\alpha$ and $\beta$ denote the hyper-parameters in the encoder and decoder. In order to achieve the latent representation of input and reconstructed output, downsampling and upsampling operations are required. Since the CAE network constructed in this work is not deep, consecutive sampling operations can be performed while maintaining network performance. During the encoding process, different kernels slide over the input with a stride of 2 to obtain downsampled compressed feature maps. Different from encoding, the decoding process uses transpose convolution kernels to expand one input pixel to 2 × 2 pixels to achieve upsampling. During transpose convolution, the stride and padding are set to 2 and 1, respectively. As shown in Fig. 1b, the compressed feature maps with size 16 × 16 × 8 (width × height × depth) are acquired after convolution with kernel weights of size 3 × 3 × 3 × 8 (width × height × depth × batch). Then, the reconstructed feature maps with size of 32 × 32 × 3 are formed after transpose convolution with 2 × 2 × 8 × 3 kernel weights. Eight convolution kernels are needed to extract enough effective information from the input image for subsequent reconstruction. Since there is only one transpose convolution layer, the number of kernels is set to three according to the RGB format of the input image. For the four-layer and six-layer CAE networks, the size of the convolution kernels remains unchanged while the number doubles with each convolution layer. The variation in depth and quantities of the transpose convolution kernels correspond to the convolution kernels. Consequently, the CAE networks with two-, four-, and six-layer have convolution/transpose convolution kernel elements of 216/96, 1368/608, and 5976/2656, respectively.

To train the CAE network, we utilized the Cifar-10 training dataset which comprises 50,000 color images. During training, we employed the Adam optimizer[49] and set the batch size to 16. The learning rate is decayed from 0.01 to 0.0004 over 300 epochs. To measure the distortion between the original and reconstructed images, mean square error (MSE) is used as the loss function during training. For testing, we used both the ImageNet dataset and Kodak24 dataset. Since the input size is set to 32 during training, high-resolution images during testing are split into 32 × 32 patches.

After software training, we conducted step-by-step quantization aware training to accommodate for different dimensional changes in the encoder and decoder. The learning rate is kept at a constant of 0.001. In the first step, the compressed data is quantized (6-bit) and introduced it into the training while maintaining the kernel weights in the encoder and decoder as 32-bit single-precision floating-point. Next, the weights in encoder are quantized (8-bit) and trained them while keeping the weights in the decoder (32-bit single-precision floating-point weights) and compressed data (6-bit quantized data) unchanged. Similar to the second step, the final step is to quantize and train the weights in the decoder. We conducted five-epoch training in each step to attain the optimal network performance after quantization.

During the inference process, variations that might be caused by memristor conductance fluctuation or readout operation are introduced to evaluate their impact on network performance. To add variations, the weight or compressed data is first converted into bit numbers based on the storage capacity of the memristor. The normalized σ is then added to the corresponding bit of the weight, followed by the reconversion to floating point and being used for subsequent computations.

To assess the quality of the image reconstruction, the typical PSNR metric is used. The formula of PSNR is as follows:

$$PSNR = 20\log_{10}\left(\frac{MAX_f}{\sqrt{MSE}}\right) \tag{1}$$

where $MAX_f$ is the maximum pixel value in the original image. $MSE$ represents the mean square error between the original image and reconstructed image, which can be expressed as follows:

$$MSE = \frac{1}{lmn}\sum_0^{l-1}\sum_0^{m-1}\sum_0^{n-1}\|f(l,m,n) - g(l,m,n)\|^2 \tag{2}$$

where $l$, $m$, $n$ denote the depth, length, and width of the original image, respectively. $f(\cdot)$ and $g(\cdot)$ denote the matrix data of the original and reconstructed image, respectively.

## ResNet34 for image recognition

The structure of the ResNet34 network used for recognition is similar to that in reference[50], which contains one convolution layer, sixteen residual blocks, and a fully-connected layer. Each residual block consists of two convolution layers (kernel size of 3 × 3) and a shortcut (kernel size of 1 × 1). The sixteen residual blocks are divided into four groups, in which each group contains four residual blocks. The four groups correspond to 64, 128, 256, and 512 output channels, respectively. The ResNet34 network is trained with two types of training datasets (original ImageNet dataset and reconstructed ImageNet dataset) to achieve two networks with different parameters. These two trained networks are used to test the original and reconstructed test datasets, respectively.

## System overhead evaluation

The schematic diagrams of the system architectures used for evaluation are shown in Supplementary Fig. 17. Here, we use the VisualQA v2.0 dataset[47], which is 25 GB with 265016 images, to evaluate the latency and energy. The evaluation starts from the input of the data for compression and ends with the output of the reconstructed data. In the CPU/GPU/ASIC-based processing system, the data is input into the CPU/GPU/ASIC to perform the compression computation. The

 

compressed data is fed into the hard disk drive (HDD) via the dynamic random-access memory (DRAM). For data retrieval, the stored data is fetched to CPU/GPU/ASIC via DRAM and output after the decode computation. In the memristor-based non-near-storage in-memory processing system, the data is input into the memristor-based in-memory computing bank to perform the data compression with the CAE algorithm and then stored into the HDD. The stored data is read from the HDD and fed into the in-memory computing unit for data retrieval. In our proposed system, the memristor-based in-memory computing bank compresses the input data and stores it into the memristor-based storage bank. When the data is required for retrieval, the data is read out from the memristor-based storage bank and fed into the memristor-based in-memory computing bank for decoding with the CAE algorithm. The energy consumption is the sum of the energy consumption of all components. Due to the dataset being processed in a pipeline fashion during compression/retrieval, the system's latency depends on the process that consumes the most data processing and storage time.

**Server-grade central processing unit-based processing system.** We configured a Dell PowerEdge R750 rack server with two 28-core Intel Xeon 6330 (10 nm) and sixteen 64 GB DDR4 SDRAM for latency and energy consumption evaluation. Intel® Power Gadget 3.6 software[51] is used to measure the latency and power/energy of the encoding and decoding processes. The system latency is obtained by measuring the actual running time of the program. The energy consumption is calculated as follows: thermal design power × CPU usage of executing process × system latency. The network structure of AE refers to[36], which has 4096 neurons in the input and output layers and 448 neurons in the hinder layer (compressed layer). The CAE network uses the two-layer structure demonstrated in the manuscript, which has 8 convolution kernels with size $3 \times 3 \times 3$ and 3 transpose convolution kernels with size $2 \times 2 \times 8$. Both AE and CAE networks are performed in the Pytorch framework. Since the data storage and evaluation are performed on Dell PowerEdge R750 with SEAGATE (ST4000NM00A) HDD as storage media, the parameters from the SEAGATE datasheet[52] are used for calculation. The latency and energy consumption generated by the data access, writing/reading are estimated according to the interface access speed, max sustainable transfer rate, and average operating power, which can be acquired from the datasheet of 1.5 Gb/s, 215 MB/s, and 10.77 W, respectively. Based on the above considerations and parameters, the evaluation results on latency and energy consumption of server-grade CPU-based processing system are listed in Supplementary Tables 2, 3, respectively.

**Graphics processing unit -based processing system.** We employed an NVIDIA Tesla T4 GPU (TSMC 12 nm) backed with 16 GB GDDR6 DRAM to evaluate the latency and energy consumption. The system latency is obtained by measuring the actual running time of the program. The energy consumption is calculated as follows: (program execution power - system standby power) × system latency. The CUDA Toolkit[53] is used to perform CAE models on the GPU and simultaneously test the latency and power consumption. The power statistics, including GPU and DRAM power, are measured using the nvidia-smi command in the terminal. The parameters for HDD can be referenced setting in the CPU-based processing system section. According to the above metrics, the latency and energy consumption for compressing and decompressing the VisualQA v2.0 dataset are calculated in Supplementary Table 4.

**Application-specific integrated circuit-based processing system.** The ASIC-based processing system consists of a hardware accelerator and off-chip DRAM and storage[54]. The recently reported CHIMERA accelerator (40 nm CMOS technology)[55] with 0.92 TOPS on peak performance and 2.2 TOPS/W on energy efficiency is adopted for the

evaluation. DDR4 DRAM with 8 memory controllers and 64-bit channel per controller is utilized as external memory, which reported 45 pJ/bit on read/write energy and 65 ns/60 ns on read/write latency[56]. For data compression/retrieval, the latency/energy consumption can be obtained by dividing the number of operations by peak performance/energy efficiency. For the DRAM fetching process, the latency is calculated by multiplying the read/write latency per operation and the operation numbers, and the energy consumption can be achieved by multiplying the energy consumption per bit and the data size. For the VisualQA v2.0 dataset, the total number of operations during encoding/decoding is 900 G/400 G for the CAE network. The transferred data size after compression is 100 Gb. The parameters for HDD can be referenced in the GPU-based processing system section. Based on the above parameters, the evaluation results on latency and energy consumption can be estimated, as listed in Supplementary Table 5. Although there are other ASIC and DRAM with relatively high performance, the improvement in latency and energy consumption of the compressed system based on ASIC is slight since the bottleneck is the data transfer and storage.

**Memristor-based near-storage in-memory processing system.** The architecture of the memristor-based near-storage in-memory processing system consists of 2133 storage banks and 8 in-memory computing banks for VisualQA v2.0 dataset compression and storage, with architecture shown in Supplementary Fig. 1. Each computing and storage bank include 16 cores. The size of the memristor array is set to be 1 Mb ($1024 \times 1024$)[57] in the storage core, and that in the computing core remains 4 kb ($256 \times 16$) as discussed in the manuscript. The latency and energy consumption are dominated by the memristor array, analog-to-digital converter (ADC), digital-to-analog converter (DAC), and buffer, mainly considered during evaluation[58–61]. During the compression process, the weights of the two-layer CAE network can be completely mapped into one computing core, so one bank can map 16 sets of weights and realize parallel computing. The dataset is compressed to $16.67 \times 10^9$ pixels with 6-bit accuracy for each pixel by the two-layer CAE network. In this case, the compressed data requires about 2133 storage banks for data storage, and different storage banks can perform parallel writing or reading of data. Accordingly, 8 computing banks are integrated during evaluation to boost speed.

All the circuit components in the near-storage in-memory processing system are evaluated under 65 nm technology node. A 0.2 V read voltage with a 20 ns pulse width is adopted for the single-bit calculation of the memristor array. The mean weight value, extracted from the conductance distribution of all cells in the weight matrix, represents the weight of each cell, is used for calculating the energy consumption. Similarly, the conductance value used for data writing and reading is extracted from the conductance distribution of the compressed data matrix. For data writing, the pulse numbers used for multi-bit memristor programming are calculated according to the total programming pulses (see Supplementary Fig. 20). We use CACTI[62] to evaluate the latency and energy consumption of the buffer, which is designed to be 64 KB in size and 2 kb in bandwidth to allow for parallel computing of all 16 cores in the computing bank. The ADC in[63] is adopted, which reported an 8-bit resolution with a sampling rate of 450 MS/s and an active area of 0.035 mm². 16 SLs are operating in parallel with one ADC on each. We use the DAC data from[64], which reported 136 mW for digital/analogue conversion under 20 GS/s. All the parameters are summarized in Supplementary Table 6. According to these metrics, the latency and energy consumption for compressing and decompressing the VisualQA v2.0 dataset are calculated in Supplementary Table 7.

**Memristor-based non-near-storage in-memory processing system.** Since the non-near-storage in-memory processing approach combines memristor-based in-memory computing bank and HDD storage, the

latency and energy consumption can be calculated based on available device and circuit metrics setting in the above discussion.

## Storage density evaluation

During evaluation, the compression ratio CR is defined as the ratio of input data bit numbers to compressed data bit numbers, which is expressed as:

$$CR = \frac{W_0 \times H_0 \times C_0 \times B_0}{W_r \times H_r \times C_r \times B_r} \quad (3)$$

where $W$ and $H$ represent the width and height of the images, respectively. $C$ is the channel of the image, which is 3 for original color maps (red, green, and blue). $B$ is the bit number of each pixel. Subscript $O$ and $r$ denote the original and compressed feature map, respectively.

The storage density improvement is defined as the ratio of the total area overhead used for compression/retrieval and data storage. We consider the area of memristor array not only for data storage but also for all encoding/decoding processing as well as corresponding peripheral circuitry in cores and banks during evaluation. We still use the VisualQA v2.0 dataset to evaluate the area. The architecture of the proposed system is shown in Supplementary Fig. 1. The system consists of 8 in-memory computing banks, and the number of storage banks depends on the storage capacity of the memristors. The area parameters of the buffer, ADC, DAC, and the memristor array in the computing bank can be referred to Supplementary Table 6. In the computing core, the register is designed based on D flip flop (DFF)[65], which has 8-bit for each WL. The MUX and sample and hold (S & H) circuits performance are simulated and optimized in Cadence. The shift & add circuit is designed to realize 16-bit shift addition based on the adder and DFF acquired in references[65,66]. The area metrics of a computing core are shown in Supplementary Table 1. In the storage core, the area of one 1T1M cell is set to 12 $F^2$ during evaluation, where F represents the feature size. In this case, the area of WL, BL, and SL drivers and MUX, composed of inverters or gate circuits, can be estimated based on the feature size, as shown in Supplementary Table 8. The peripheral circuits, including DAC, S & H, ADC, and shift & add, are located in the storage bank outside the core. The original VisualQA v2.0 dataset requires 25 GB storage space for data storage, which needs the HDD disk and its peripheral circuitry area of about 7194 mm$^2$ for storage[52]. The areas of CPU and DRAM for processing are estimated to be 2.48 and 84.23 mm$^2$[67,68].

## Data availability

The datasets used for evaluation and recognition in this study are publicly available. The source data underlying the figures in the main manuscript are provided as Source Data file. Additional data supporting the findings of this study are available from the corresponding authors upon request. Source data are provided with this paper.

## Code availability

The code that supports the results within this paper and the other findings of this study are available from the corresponding authors upon request.

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

## Acknowledgements

We gratefully appreciate the invaluable suggestions provided by Prof. H. Qian, Prof. H. Q. Wu, and Prof. B. Gao from Tsinghua University. This work was supported in part by the National Sci-Tech Innovation 2030 under Grant 2021ZD0201200 (L. L.), the National Natural Science Foundation of China 62022006 (P. H.), 61834001 (L. L.), and the 111 Project B18001 (P. H.).

## Author contributions

P.H. and Y.F. conceived the concept and designed the experiments. Y.F. and Y.Z. contributed to the fabrication of memristor array and performed the measurements. Y.F., Y.Z. and Z.Z. contributed to the neural network simulation. Y.F., Z.Z., and P.H. analyzed the data. Y.F.,P.H.,L.L.,X.L., and

J.K. wrote the manuscript. All authors contributed to the results analysis and commented on the manuscript. P.H. and L.L. supervised the project.

## Competing interests

The authors declare no competing interests.
