## [Peer Review File · Nature Communications]

REVIEWER COMMENTS

Reviewer #1 (Remarks to the Author):

The manuscript "Memristor-based high-density storage system with deep learning-based image compression network" by Y. Feng et al., presents an intriguing approach of using a memristor-based image storage mechanism in conjunction with a convolutional neural network-based auto-encoder/decoder for image compression and reconstruction. The study includes experimental demonstrations on a 256 x 16 1T1R crossbar array and explores various software-hardware co-design methodologies to overcome device non-idealities. The authors claim a significant 6x (600%) increase in storage density, which is commendable.

The memristor, or resistive random-access memory (RRAM), was initially developed as non-volatile memory and later incorporated into in-memory computing. This approach combines memory and computing functionalities, unifying in-memory computing and near-storage computing. The idea is intriguing, and the results have been promising. This method is particularly suited to storing analog data, where a certain degree of data loss (lossy compression) is acceptable because all computation and data storage are performed in the analog domain. Therefore, the concept proposed in this work holds practical promise.

Still, there are some points of concern regarding the implementation details and a requirement for a more comprehensive comparison with competitive technologies. Here are my suggestions for improvement:

1. The reviewer's opinion is that the true potential of this method may reside more in the speed and energy efficiency of the data compression and reconstruction process, rather than solely in the compression ratio demonstrated in this preliminary study. This is because the combination of near-storage processing and in-memory processing significantly reduces data transfer. Accordingly, it would be beneficial to conduct a benchmark comparison with systems utilizing competitive technologies (such as JPEG / CAE implemented in non-near-memory approaches) to gain a fair understanding of latency and energy consumption during data compression/retrieval.
2. Secondly, it is suggested to provide more clarifications on how 6x storage density is achieved. It appears that the proclaimed 6x increase in storage density stems from the combination of the memristor device's multi-level (3-bit) storage capability and the compression ratio of 2x. Both components could be substituted with alternate technologies. For instance, the storage component could be replaced with multilevel Flash to achieve a higher storage density. The encoding/decoding component could be implemented in software or other accelerators which wouldn't impact the reported density improvement, given that the area used for encoding/decoding isn't factored into this equation. The compression ratio showcased in this proof-of-concept experiment doesn't inspire awe as yet. However, the reviewer also agrees that it holds potential for enhancement with a larger network or a more capable device. The reviewer concurs that a high compression ratio doesn't necessarily have to be demonstrated in this study as long as scalability for the future can be demonstrated. Still, the compression ratio can be more fairly discussed, for example taking into consideration the area needed for all encoding / decoding arrays and peripheral circuits (bit-slicing, shift-and-add, analog-digital signal conversions, etc.).

Additional minor suggestions:

1. Related to the above comment: Line 331: Could the authors provide the formula used for calculating the compression ratio? Considering the number of features, from 32x32x3 to 16x16x8, it appears to be a compression ratio of 3:2 rather than 2.
2. The reasoning behind choosing 2-bit weight and 3-bit data storage, while Fig. 2c shows a reliable 4-bit with 16 levels, should be explained.
3. The method of implementing analog input in the 1T1R memristor crossbars, where inputs can only be binary, should be clarified. If a bit-slicing encoding technique is employed, what are the potential latency overhead and complexity of peripheral circuitry?
4. The array's size (256 x 16) deviates from a square shape; it would be helpful to provide the rationale behind choosing this size.
5. The equivalent transformation of transpose convolution could use more explanation, such as mathematical expressions or code, for better understanding.

6. The digital readout method appears common for multi-level digital memory systems. The readout is quantized naturally when the ADC precision is limited. Please highlight the novelty in this approach compared to others.

7. The "visualization degree" seems to be a subjective measure of image quality. The use of more objective, quantitative metrics like PSNR is encouraged. CIFAR-10 accuracy might not be the best choice unless its relevance to the use-case scenario is explained.

8. Fig. 3c: Please explain why PSNR decreases with more codec layers. Is it due to higher compression ratio leading to more information loss?

The overall concept and experimental demonstration of your work are very promising. I hope that addressing the above points will improve the manuscript and its potential impact on the scientific community.

Reviewer #2 (Remarks to the Author):

The authors present a storage system based on memristor which uses an autoencoder for compressing data. Both storage and the autoencoder stages are realized with memristors. The paper is well-written, and the topic is relatively new. Experiments are provided to demonstrate the concept in practice. However, some issues should be addressed before considering publication:

1. The mapping of high precision weights is not clear in Supplementary Fig. 1. Why for example the weight $W_{\{1,1\}} = 4$ is converted to [0010 - 0000] and not [0100 - 0000] since the number 4 converted in binary with 4 bits is 0100, or $0100_2 = 4_{10}$?

2. The precision of VMM from the Supplementary Fig. 4 is not clear. I suggest performing a correlation plot of the SW (ideal) VMM and the measured experimental VMM.

3. Line 212, where do the 'magic' numbers of 8 bits for the weights and 6 bits for the data come from? Did the authors perform a design space exploration with different precisions?

4. Line 218, Fig. 3b does not show performance improvement after retraining. In general, references to figure 3 seem odd

5. Line 233, is the digital readout proposed the same proposed in "Zidan, M.A., Jeong, Y., Lee, J. et al. A general memristor-based partial differential equation solver. Nat Electron 1, 411-420 (2018). <https://doi.org/10.1038/s41928-018-0100-6>" (see SI Fig 6)? And if not, what are the differences?

6. Generally, it is not clear if the storage system was tested with a class previously unseen by the autoencoder. What happens if something new is seen? Is it able to reconstruct it or a more complicated network is needed? I think this is critical, a storage system should be independent of the 'class' of the data that is being stored

7. Line 362, a simulation of what would happen using memristors with more conductance states should be done to motivate the statement

8. Compared with ref 43 and 36 it seems that the novelty of this work is an experimental verification of the concept. While the effort is significant and important, could the authors clarify the main conceptual novelty (not technical, such as autoencoder vs convolutional autoencoder) of this work?

Minor comments:

1. Line 168, the sentence is a bit confusing, it reads better as "a 4kb (256x16) crossbar arrays with TiN/TaOx/HfOx/TiN structure is fabricated on top of the CMOS circuits"

Reviewer #3 (Remarks to the Author):

This work presents an experimental demonstration of image compression with multi-bit resistive memories with packaged chip measurements. The idea essentially combines two aspects to achieve high storage density of compressed data: convolutional autoencoder network as the compression technique and utilizing multi-level cell capability of metal-oxide resistive memories. This is a comprehensive study of the feasibility of multi-bit resistive RAM for competing with existing high-density data storage solutions. However, the reviewer found that the work is not sufficiently novel to justify publication in this journal at the current stage, as both aspects were

well studied in the prior works and this paper fails to show a significant leap towards more possibilities or new applications. Specific comments are listed below that might help with improving the overall quality for submission to other venues.

1. It's a bit misleading to readers when the authors use the word "deep learning" in the title with a two-layer shallow network (CAE).
2. Multi-bit conductance readout and conversion to analog width-modulated pulses are not discussed clearly. Such conversion overhead might be high to generate accurate, analog pulses.
3. With all the overhead considered, it is necessary to benchmark the RRAM-based solution with conventional digital solutions to show how the multi-bit capability can boost overall efficiency.

RESPONSE TO REVIEWERS' COMMENTS

Response to the Reviewer 1

Overall Comment:

The manuscript "Memristor-based high-density storage system with deep learning-based image compression network" by Y. Feng et al., presents an intriguing approach of using a memristor-based image storage mechanism in conjunction with a convolutional neural network-based auto-encoder/decoder for image compression and reconstruction. The study includes experimental demonstrations on a 256 x 16 1T1R crossbar array and explores various software-hardware co-design methodologies to overcome device non-idealities. The authors claim a significant 6x (600%) increase in storage density, which is commendable.

The memristor, or resistive random-access memory (RRAM), was initially developed as non-volatile memory and later incorporated into in-memory computing. This approach combines memory and computing functionalities, unifying in-memory computing and near-storage computing. The idea is intriguing, and the results have been promising. This method is particularly suited to storing analog data, where a certain degree of data loss (lossy compression) is acceptable because all computation and data storage are performed in the analog domain. Therefore, the concept proposed in this work holds practical promise.

Still, there are some points of concern regarding the implementation details and a requirement for a more comprehensive comparison with competitive technologies. Here are my suggestions for improvement:

Response:

Many thanks for the Reviewer's time reviewing the manuscript and encouraging comments on the merits. We also appreciate the Reviewer's insightful and detailed suggestions, which greatly help us improve the quality of the manuscript substantially. We have carefully studied all comments from the Reviewer and revised our manuscript accordingly. Significantly, according to the Reviewer's suggestions, we have adopted the 4-bit memristors for the hardware implementation of the proposed system and compared it with other systems utilizing competitive technologies. The point-by-point responses to the comments and the corresponding revisions in the manuscript are as follows. We hope that the responses have fully addressed all of the concerns.

Comment #1:

The reviewer's opinion is that the true potential of this method may reside more in the

speed and energy efficiency of the data compression and reconstruction process, rather than solely in the compression ratio demonstrated in this preliminary study. This is because the combination of near-storage processing and in-memory processing significantly reduces data transfer. Accordingly, it would be beneficial to conduct a benchmark comparison with systems utilizing competitive technologies (such as JPEG / CAE implemented in non-near-memory approaches) to gain a fair understanding of latency and energy consumption during data compression/retrieval.

Response:

We gratefully appreciate the Reviewer’s constructive suggestion. We agree with the Reviewer that the potential of the proposed method also resides in the speed and energy efficiency of the data compression and reconstruction process. As the Reviewer suggested, we compare the latency and energy consumption of the proposed system with that of other competitive technologies, including JPEG, PNG, autoencoder (AE), and convolutional autoencoder (CAE) implemented in the server-grade central processing unit (CPU)-based processing system, and CAE implemented in the non-near-storage in-memory processing system. Since compression techniques usually contain various domain transformations and complex lossless coding schemes (such as Huffman coding), in most cases, the compression is implemented based on CPU. The benchmark comparison results are shown in Fig. R1, which indicates that the proposed system reduces the latency and energy consumption by more than 20× and 180× compared with server-grade CPU-based processing system, respectively, and 5.6× and 30× compared with non-near-storage in-memory processing system, respectively. The detailed **Evaluation method** is as follows.

Fig. R1 **a**, Latency and **b**, energy comparison of server-grade CPU-based processing system, non-near-storage in-memory processing system, and near-storage in-memory processing systems.

Evaluation method

The schematic diagrams of the system architectures used for evaluation are shown in Fig. R2. Here, we use the VisualQA v2.0 dataset [1], which is 25 GB with 265016 images, to evaluate the latency and energy. The evaluation starts from the input of the

data for compression and ends with the output of the reconstructed data. In the CPU-based processing system, the data is input into the CPU, and perform the compression computation. The compressed data is fed into the hard disk drive (HDD) via the dynamic random-access memory (DRAM). For data retrieval, the stored data is fetched by the CPU via DRAM and output after the decode computation. In the memristor-based non-near-storage in-memory processing system, the data is input into the memristor-based in-memory computing bank to perform the data compression with the CAE algorithm and then stored into the HDD. The stored data is read from the HDD and fed into the in-memory computing unit for data retrieval. In our proposed system, the memristor-based in-memory computing bank compresses the input data and stores it into the memristor-based storage bank. When the data is required for retrieval, the data is read out from the memristor-based storage bank and fed into the memristor-based in-memory computing bank for decoding with the CAE algorithm. The energy consumption is the sum of the energy consumption of all components. Due to the dataset being processed in a pipeline fashion during compression / retrieval, the system's latency depends on the process that consumes the most data processing and storage time.

Fig. R2 Schematic diagram of **a**, CPU-based processing system, **b**, Memristor-based non-near-storage in-memory processing system, and **c**, Memristor-based near-storage in-memory processing system architectures.

Server-grade CPU-based processing system:

We configured a Dell PowerEdge R750 rack server with two 28-core Intel Xeon 6330 (10nm) and sixteen 64 GB DDR4 SDRAM for latency and energy consumption evaluation. Intel® Power Gadget 3.6 software [2] is used to measure the latency and power / energy of the encoding and decoding processes. The system latency is obtained

by measuring the actual running time of the program. The energy consumption is calculated as follows: thermal design power \times CPU usage of executing process \times system latency. The network structure of AE refers to [3], which has 4096 neurons in the input and output layers and 448 neurons in the hinder layer (compressed layer). The CAE network uses the two-layer structure demonstrated in the manuscript, which has 8 convolution kernels with size $3 \times 3 \times 3$ and 3 transpose convolution kernels with size $2 \times 2 \times 8$. Both AE and CAE networks are performed in the Pytorch framework. Since the data storage and evaluation are performed on Dell PowerEdge R750 with SEAGATE (ST4000NM00A) HDD as storage media, the parameters from the SEAGATE datasheet [4] are used for calculation. The latency and energy consumption generated by the data access, writing / reading are estimated according to the interface access speed, max sustainable transfer rate, and average operating power, which can be acquired from the datasheet of 1.5 Gb/s, 215 MB/s, and 10.77 W, respectively. Based on the above considerations and parameters, the evaluation results on latency and energy consumption of server-grade CPU-based processing system are listed in Table R1 and Table R2, respectively.

Table R1. Latency of server-grade CPU-based processing system.

Compression technique	Process	CPU + DRAM (s)	Storage (s)
JPEG	Encode	535.98	42.12
	Decode	213.94	42.12
PNG	Encode	2072.07	240.33
	Decode	344.28	240.33
AE	Encode	1066.87	27.29
	Decode	1028.09	27.29
CAE	Encode	2267.26	124.8
	Decode	1061.15	124.8

Table R2. Energy consumption of server-grade CPU-based processing system.

Compression technique	Process	CPU + DRAM (J)	Storage (J)
JPEG	Encode	3884.74	212.99
	Decode	1550.71	212.99
PNG	Encode	15171.14	1210.6
	Decode	2520.63	1210.6

AE	Encode	43491.58	136.87
	Decode	41910.66	136.87
CAE	Encode	92808.36	626
	Decode	43437.22	626

Memristor-based near-storage in-memory processing system:

The architecture of the memristor-based near-storage in-memory processing system consists of 2133 storage banks and 8 in-memory computing banks for VisualQA v2.0 dataset compression and storage, as shown in Fig. R3. Each computing and storage bank include 16 cores. The size of the memristor array is set to be 1-Mb (1024×1024) [5] in the storage core, and that in the computing core remains 4-kb (256×16) as discussed in the manuscript. The latency and energy consumption are dominated by memristor array, analogue-to-digital converter (ADC), digital-to-analogue converter (DAC), and buffer, which are mainly considered during evaluation [6-9]. During the compression process, the weights of the two-layer CAE network can be completely mapped into one computing core, so one bank can map 16 sets of weights and realize parallel computing. The dataset is compressed to 16.67×10^9 pixels with 6-bit accuracy for each pixel by the two-layer CAE network. In this case, the compressed data requires about 2133 storage banks for data storage, and different storage banks can perform parallel writing or reading of data. Accordingly, 8 computing banks are integrated during evaluation to boost speed.

Fig. R3 Diagram of memristor-based near-storage in-memory processing architecture. **a**, Computing bank and a memristor-based computing core. **b**, Storage bank and internal structure of a storage core.

All the circuit components in the near-storage in-memory processing system are evaluated under 65 nm technology node. A 0.2 V read voltage with a 20 ns pulse width is adopted for the single-bit calculation of the memristor array. The mean weight value, extracted from the conductance distribution of all cells in the weight matrix, represents the weight of each cell is used for calculating the energy consumption. Similarly, the conductance value used for data writing and reading is extracted from the conductance

distribution of the compressed data matrix. For data writing, the pulse numbers used for multi-bit memristor programming are calculated according to the total programming pulses (Fig. R4). We use CACTI [10] to evaluate the latency and energy consumption of the buffer, which is designed to be 64 KB in size and 2-kb in bandwidth to allow for parallel computing of all 16 cores in the computing bank. The ADC in [11] is adopted, which reported an 8-bit resolution with a sampling rate of 450 MS/s and an active area of 0.035 mm². 16 source lines (SLs) are operating in parallel with one ADC on each. We use the DAC data from [12], which reported 136 mW for digital / analog conversion under 20 GS/s. All the parameters are summarized in Table R3. According to these metrics, the latency and energy consumption for compressing and decompressing the VisualQA v2.0 dataset are calculated in Table R4.

Fig. R4 Statistical pulse numbers to program 4-bit memristor. Programming pulses of 100 cells are counted for each conductance state.

Table R3. Summary of metrics of key circuit modules in memristor-based computing bank used for evaluation.

Module	Energy	Latency	Area (μm^2)
ADC	14.3 pJ/bit	2.2 ns/op	35000
DAC	6.8 pJ/conversion	50 ps/op	72000
Buffer	0.38 pJ/bit	1.04 ns/access	1360000
Array	9.2 fJ/cell (encoding) 8.2 fJ/cell (decoding) 17 fJ/cell (reading) 25 pJ/cell (writing)	20 ns/pulse @ reading 100 ns/pulse @ writing	9646 @ 4-kb

Table R4. Evaluation results of latency and energy during data compression and retrieval with memristor-based near-storage in-memory processing system.

Performance	Process	Buffer	Array	ADC	Data write/access	Overall
Latency (s)	Encoding	0.27	10.41	0.57	22.14	22.14
	Decoding	0.3	11.71	0.64	0.67	11.71
Energy (J)	Encoding	1.37	0.13	7.63	11.66	20.79
	Decoding	0.46	0.04	8.58	0.7	9.78

Memristor-based non-near-storage in-memory processing system:

Since the non-near-storage in-memory processing approach combines memristor-based in-memory computing bank and HDD storage, the latency and energy consumption can be calculated based on available device and circuit metrics setting in the above discussion.

The following revisions have been made in the main text and supplementary information.

- 1) Page 15, line 430, lines 439-443. The benchmark comparison results of latency and energy consumption are added into Fig. 5 as Fig. 5d and 5e. The caption of Fig. 5 is modified accordingly.
- 2) The architectures of different storage systems and memristor-based computing / storage bank are added into supplementary information as Supplementary Fig. 16 and Fig. 1, respectively.
- 3) Table R1, Table R2, Table R3, and Table R4 are replenished into supplementary information as Supplementary Table 2, Table 3, Table 4, and Table 5, respectively.
- 4) The data of statistical programming pulse numbers is replenished into supplementary information as Supplementary Fig. 19.
- 5) Pages 14, lines 395-412. The evaluation results are added into manuscript, which is written as: “We compared the latency and energy consumption of the proposed system with that of other competitive technologies, including JPEG, PNG, autoencoder (AE), and CAE implemented in the server-grade CPU-based processing system, and CAE implemented in the non-near-storage in-memory processing system (see Supplementary Fig. 16 and Methods). Since compression techniques usually contain various domain transformations and complex lossless coding schemes (such as Huffman coding), in most cases, the compression is implemented based on the CPU. As illustrated in Fig. 5d and 5e, the near-storage in-memory processing system effectively reduces the latency and energy consumption by 20× and 180× compared with server-grade CPU-based processing systems, respectively, and 5.6× and 30× compared with non-near-storage in-memory processing system, respectively. The significant improvements in energy

efficiency can be mainly attributed to the efficient memristor-based in-memory computing technique and reduction in data transfer between computing unit and storage, manifesting the proposed system's considerable potential and energy efficiency advantages in data compression and retrieval. In addition, it can also be inferred that the latency of the compression/reconstruction process is primarily consumed by both data computing and storage. While the storage process mainly dominates energy consumption".

- 6) The contents in Abstract is modified accordingly.

Page 1, lines 13-16. "Here, we propose a memristor-based storage system with an integrated near-storage in-memory computing-based convolutional autoencoder compression network to boost the energy efficiency and speed of the image compression / retrieval and improve the storage density".

Page 1, lines 24-27. "Benchmark comparison results show that the 4-bit memristor-based storage system could reduce the latency and energy consumption by over 20× and 180× compared to the server-grade central processing unit (CPU)-based system and improve the storage density by more than 3 times".

- 7) The last paragraph of Introduction is modified accordingly.

Page 3, lines 66-68. "This study proposes a memristor-based storage system integrated with the near-storage in-memory computing for data compression / decompression and multi-bit storage banks".

Page 3, lines 78-81. "The evaluation results indicate that the proposed system effectively reduces latency and energy consumption by more than 20× and 180× compared with the server-grade central processing unit (CPU)-based processing system and improves storage density by more than 3 times".

- 8) The contents in Discussion is modified accordingly.

Page 16, lines 447-450. "In this study, we proposed a memristor-based storage system that integrates a deep image compression network for image data storage, which is implemented based on a near-storage in-memory processing architecture with high efficiency and storage density".

Page 16, lines 457-460. "The benchmark comparison results illustrate that the memristor-based near-storage in-memory processing system can achieve reductions of over 20× in latency and 180× in energy consumption compared to the server-grade CPU-based processing system".

- 9) Pages 20-22, lines 581-660. We added a new section named "System overhead evaluation" into Methods, the evaluation details on latency and energy consumption are added into this section.

- 10) Page 5, line 147. The phrase "Neural processing and memory unit" in Fig. 1c is changed to "Neural processing and storage unit", to more accurately describe the

functionalities of the unit.

Comment #2:

Secondly, it is suggested to provide more clarifications on how 6x storage density is achieved. It appears that the proclaimed 6x increase in storage density stems from the combination of the memristor device's multi-level (3-bit) storage capability and the compression ratio of 2x. Both components could be substituted with alternate technologies. For instance, the storage component could be replaced with multilevel Flash to achieve a higher storage density. The encoding/decoding component could be implemented in software or other accelerators which wouldn't impact the reported density improvement, given that the area used for encoding/decoding isn't factored into this equation. The compression ratio showcased in this proof-of-concept experiment doesn't inspire awe as yet. However, the reviewer also agrees that it holds potential for enhancement with a larger network or a more capable device. The reviewer concurs that a high compression ratio doesn't necessarily have to be demonstrated in this study as long as scalability for the future can be demonstrated. Still, the compression ratio can be more fairly discussed, for example taking into consideration the area needed for all encoding / decoding arrays and peripheral circuits (bit-slicing, shift-and-add, analog-digital signal conversions, etc.).

Response:

Many thanks for the Reviewer's valuable comment. In the previous manuscript, the storage density improvement = cell number_(original) / cell number_(compressed), where cell number = length × width × depth × bit per pixel / storage capacity per cell. Since one magnetic grain (or magnetic particle) can only store 1 bit, the HDD storage capacity per cell is 1 bit. In this case, the number of cells required to store an RGB image with size 32×32 is calculated as cell number_(original) = 32×32×3×8/1 = 24576. For compressed data storage, we use memristor arrays with 3-bit storage capacity, and the data of each pixel after compression is quantized to 6-bit. In this case, the number of cells required to store an RGB image with size 32×32 is calculated as cell number_(compressed) = 16×16×8×6/3 = 4096. The storage density improvement is calculated as 24576 / 4096 = 6.

In the revised manuscript, we consider all the areas needed for encoding / decoding, including the memristor arrays and peripheral circuitry, for a fairer discussion on the storage density according to the Reviewer's suggestion. Fig. R5 shows the evaluated result, which implies that the storage density increases by more than 3 times. As suggested by the Reviewer, we have also added the discussion about the potential improvement in storage density based on the proof-of-concept experiment to the revised manuscript. The evaluation results indicate that a maximum multiple of 18× is achieved

under the condition of a six-layer CAE network and 8-bit memristor [13], as shown in Fig. R5. The density can be further improved using the deep learning-based compression technique with a higher compression ratio or the devices with more conductance states. It should be noted that the scaling down of the technology could offer excellent potential for further improving the storage density, considering the evaluation is under 65nm technology. The detailed **Evaluation method** is as follows.

Fig. R5 Area comparison of server-grade CPU-based processing system and proposed near-storage in-memory processing system for storing VisualQA v2.0 dataset. Different codec layers and memristor storage capacities are used in proposed storage system. The area of peripheral circuitry used for encoding / decoding and storage are taken into consideration.

Evaluation method

Firstly, it must be clear that in the revised manuscript, the storage density improvement is defined as the ratio of the total area overhead used for compression / retrieval and data storage. We consider the area of memristor array not only for data storage but also for all encoding / decoding processing as well as corresponding peripheral circuitry in cores and banks during evaluation. We still use the VisualQA v2.0 dataset [5] to evaluate the area. The architecture of the proposed system is shown in Fig. R2c and Fig. R3. The system consists of 8 in-memory computing banks, and the number of storage banks depends on the storage capacity of the memristors. The area parameters of the buffer, ADC, DAC, and the memristor array in the computing bank can be referred to Table R3. In the computing core, the register is designed based on D flip flop (DFF) [14], which has 8-bit for each word line (WL). The MUX and sample and hold (S & H) circuits performance are simulated and optimized in Cadence. The shift & add circuit is designed to realize 16-bit shift addition based on the adder and DFF acquired in references [14,15]. The area metrics of a computing core are shown in Table R5. In the

storage core, the area of one one-transistor one-memristor (1T1M) cell is set to $12F^2$ during evaluation, where F represents the feature size. In this case, the area of WL, bit line (BL), and SL drivers and MUX, composed of inverters or gate circuits, can be estimated based on the feature size, as shown in Table R6. The peripheral circuits, including DAC, S & H, ADC, and shift & add, are located in the storage bank outside the core. The original VisualQA v2.0 dataset requires 25 GB storage space for data storage, which needs the HDD disk and its peripheral circuitry area of about 7194 mm^2 for storage [4]. The areas of CPU and DRAM for processing are estimated to be 2.48 and 84.23 mm^2 [16,17].

Table R5. Area metrics of each circuit components in one computing core with 4-kb (256×16) memristor array.

Module	Area (μm^2)
Register	22856
WL-MUX	21504
Array	9646
SL/BL-MUX	6208
S & H	2080
ADC	560000
Shift & adder	185600

Table R6. Area metrics of each circuit components in one storage core with 1-Mb (1024×1024) memristor array.

Module	Area (μm^2)
WL driver	155.6
Array	53162
SL/BL driver	413.6
SL/BL-MUX	2998.6

The following revisions have been made in the manuscript.

- 1) Pages 22-23, lines 661-690. We added a new section named “Storage density evaluation” in the Methods of the revised manuscript. The definitions of compression ratio and storage density improvement as well as the evaluation details on storage density are elaborated in this section.
- 2) Page 15, line 430, lines 433-437. The data in Fig. 5b for storage density evaluation and corresponding caption are updated.
- 3) Table R5 and Table R6 used for area and storage density evaluation are replenished into supplementary information as Supplementary Table 1 and Table 6, respectively.

- 4) More detailed modifications and discussions on storage density have been made and added in the manuscript. Page 13, lines 366-372. “To more fairly discuss the area overhead, the areas, including encoding/decoding, storage arrays, and peripheral circuitry used for compression/retrieval, are all considered during evaluation (see Methods). In this case, the storage density improves by a factor of 3 when using the constructed system in our demonstration (two-layer CAE network with 4-bit memristor), and a maximum storage density improvement of 18 times can be achieved if a six-layer CAE network and 8-bit memristor arrays [48] are adopted”.
- 5) The contents related to storage density improvement in Abstract, Introduction, and Discussion are modified accordingly.
Page 1, lines 24-27. “Benchmark comparison results show that the 4-bit memristor-based storage system could reduce the latency and energy consumption by over 20× and 180× compared to the server-grade central processing unit (CPU)-based system and improve the storage density by more than 3 times”.
Page 3, lines 78-81. “The evaluation results indicate that the proposed system effectively reduces latency and energy consumption by more than 20× and 180× compared with the server-grade central processing unit (CPU)-based processing system and improves storage density by more than 3 times”.
Page 16, lines 460-462. “Considering the areas of encoding / decoding array, storage array, and peripheral circuitry, the storage density is increased by a factor of 3. It can be further improved to 18 times when employing a six-layer CAE network with 8-bit memristors”.
- 6) We delete Supplementary Fig. 10 in the previous supplementary information since the hardware overhead is re-evaluated based on the area of each circuit component. Corresponding statements in the manuscript is also removed.

Additional minor suggestions:

Comment #1:

Related to the above comment: Line 331: Could the authors provide the formula used for calculating the compression ratio? Considering the number of features, from 32x32x3 to 16x16x8, it appears to be a compression ratio of 3:2 rather than 2.

Response:

We are sorry for the unclear illustration. The compression ratio CR is calculated as:

$$(1)$$

where W and H represent the width and height of the image, respectively. C is the channel of the image, which is 3 for original color maps (red, green, and blue). B is the bit number of each pixel. Subscript 0 and r denote the original image and compressed feature map, respectively. The input feature of $32 \times 32 \times 3 \times 8$ is compressed to $16 \times 16 \times 8 \times 6$. So, the compression ratio is 2.

To better understand the compression ratio, we have made following revisions.

- 1) Page 22, lines 662-668. The definition and equation of compression ratio are supplemented into the manuscript.
- 2) Page 12, lines 338-341. The statement of how to achieve $2 \times$ compression ratio is rewritten to be clearer. In caption of Fig. 4, the sentence “The resolution of the compressed feature map ... compression ratio of 2 is achieved with the demonstrated system” is rewritten as “**The resolution and pixel depth of the compressed 8 feature maps are 16×16 and 6-bit, respectively. Since the original image is 32×32 with 8-bit for 3 color channels, as a result, a compression ratio of 2 is achieved with the demonstrated system (see Methods)**”.

Comment #2:

The reasoning behind choosing 2-bit weight and 3-bit data storage, while Fig. 2c shows a reliable 4-bit with 16 levels, should be explained.

Response:

We sincerely apologize for adopting different bits for weight, data storage, and programming, which led to unnecessary confusion. According to the Reviewer’s comment, we demonstrate the near-storage in-memory processing system uniformly using 4-bit memristor cells, as shown in Fig. R6. It can be found that the peak signal-to-noise ratio (PSNR) is almost the same as in previous demonstrated result. All related data is updated in the revised manuscript according to the Reviewer’s comment.

Fig. R6 Hardware implementation of the proposed near-storage in-memory processing system with 4-bit memristors.

The following revisions have been made in the manuscript.

Page 12, line 330. The hardware demonstration of the storage system shown in Fig. 4 is updated by using 4-bit memristor for weight mapping and compressed data storage. Corresponding descriptions in manuscript are also modified.

Comment #3:

The method of implementing analog input in the 1T1R memristor crossbars, where inputs can only be binary, should be clarified. If a bit-slicing encoding technique is employed, what are the potential latency overhead and complexity of peripheral circuitry?

Response:

Yes, we employ the widely used bit-slicing technique for implementing the analogue input in the 1T1M array.

Since the computation is performed bit-by-bit, the latency (T) increases with the bit number of inputs. It can be expressed as $T = T_0 \times N$, where T_0 represents the latency for processing 1-bit input data, and N is the bit number of inputs. A shift & add circuit module is needed for peripheral circuitry design when the bit-slicing encoding technique is employed. The area ratio of shift & add circuits in an entire in-memory computing bank can be observed in Fig. R7. The bit number of shift & add circuit is set to twice the input bit number so that the carry will not be lost during calculation. The evaluation method and parameters are the same as the response in the main comment

#2. The result indicates that the shift & add circuit cost about 20% of the area when the input is 8-bit. Although the bit-slicing encoding technique will lead to relatively large latency and area, the computing accuracy and the PSNR performance would be relatively high.

Fig. R7 Area ratio of shift & add circuits in the entire in-memory computing bank and PSNR value with respect to different bit number of inputs.

To better illustrate the hardware implementation of the near-storage in-memory processing system, we have added explanations as follows.

- 1) The detailed discussion on bit-slicing encoding scheme is added to supplementary information as Supplementary Note I. Bit-slicing encoding technique.
- 2) Page 4, lines 115-119. We added the introduction of bit-slicing encoding technique into the manuscript. The sentence “To perform convolution or ... memristor array in time division” is rewritten as “We adopted the bit-slicing encoding technique for multi-bit input of the memristor array (see Supplementary Note I). This technique first involves converting the input into a sequence of binary voltage pulses, which are then flattened before inputting into the word lines (WLs) of the memristor array using time division”.
- 3) Page 11, lines 314-315. We added sentence “Since the input is converted into bits from MSB to LSB by bit-slicing encoding”.

Comment #4:

The array's size (256 x 16) deviates from a square shape; it would be helpful to provide the rationale behind choosing this size.

Response:

Thanks for the Reviewer’s comment. The reason that we designed the 1T1M array size to be 256×16 instead of square shape (64×64) is mainly to reduce the area overhead of peripheral circuitry. The detailed explanations are as follows.

- (1) Fewer columns’ design means reduced number of SL / BL-MUX, ADC, S & H, and

shift & add modules in peripheral circuitry of in-memory computing core if the size of the 1T1M array is fixed. Although it will increase the number of WL-MUX and register modules, the area overhead would decrease since the area of the circuit connected to SL / BL is more dominant than the area of the circuit connected to WL. Based on the parameters of each circuit module listed in Table R3 and R5, we compared the area overhead of the peripheral circuitry under three array structures of 256×16 , 128×32 , and 64×64 , as shown in Fig. R8. The result implies that the 1T1M array with 256×16 configuration occupies the least area.

- (2) The memristor array with size 256×16 can process vector-matrix multiplication with a maximum parallelism of 256 in WL. So, the throughput is the same as the other configuration designs, even though the calculation parallelism of SL decreases.

Fig. R8 Area overhead of a computing core for different memristor array configurations. Different colors represent different circuit modules.

We have added corresponding explanations to the manuscript as follows.

- 1) The rationale and detailed explanations for selection of 1T1M array structure are added to supplementary information as Supplementary Fig. 3 and Supplementary Note II. Memristor array structure design.
- 2) Page 6, lines 162-164. We added sentence “The array design of this size is conducive to reducing the complexity and area overhead of peripheral circuitry (see Supplementary Fig. 3, Table 1, and Note II)”.

Comment #5:

The equivalent transformation of transpose convolution could use more explanation, such as mathematical expressions or code, for better understanding.

Response:

Many thanks for the Reviewer’s suggestion. Transpose convolution computing can be expressed as:

where Y and W denote the output and kernel matrix, respectively. X' is achieved by inserting zero elements between every two elements of the input matrix X , which can be written as:

(3)

We unroll the kernel matrix (2×2) as the vector (4×1) so the convolution computing is transformed as vector-matrix multiplication (VMM) as following format:

(4)

The equation (4) can be simplified as:

(5)

There are n input channels for decoding, then:

(6)

The equivalent transformation of transpose convolution is described as equation (6), which can be accelerated by the memristor-based in-memory computing core.

The following revisions have been made in the manuscript.

- 1) The detailed explanations and mathematical expressions of equivalent transformation of transpose convolution are added to supplementary information as Supplementary Note III. Equivalent transformation of transpose convolution.
- 2) Page 5, line 135; Page 10, line 273. The schematic diagram shown in Fig. 1c, 1d, and 3g are modified based on the representations of equations.

Comment #6:

The digital readout method appears common for multi-level digital memory systems. The readout is quantized naturally when the ADC precision is limited. Please highlight

the novelty in this approach compared to others.

Response:

We agree with the Reviewer's comment that the digital readout scheme is commonly used in multi-level digital memory systems. However, the analogue readout scheme is widely adopted in various compression techniques, including JPEG and deep learning-based compression networks [3,18,19]. This study found that the PSNR values would be high when adopting a digital readout scheme since the digital readout scheme can mitigate the deviations caused by the device and circuit before decompression. We have rewritten the corresponding statements and added some explanations in the revised manuscript to avoid misunderstanding.

The following revisions have been made in the manuscript.

- 1) Page 7, lines 198-200. To make the description more accurate, the sentence "we propose effective ... decompression processes" is rewritten as "**we propose effective strategies for algorithm-device codesign of decompression processes and introduce the digital readout scheme [39] into the CAE network**".
- 2) Page 9, lines 237-240. We highlighted and added the reasons and advantages of using the digital readout scheme in the manuscript, and cited relevant reference. We added sentence: "**To eliminate this deviation, we utilized a digital readout method, which appears common for multi-bit memory systems, for hardware demonstration of deep learning-based image compression technique to improve the quality of reconstructed images**".
- 3) Page 26, lines 787-788. The reference "**Zidan, M.A., Jeong, Y., Lee, J. et al. A general memristor-based partial differential equation solver. *Nat. Electron.* **1**, 411–420 (2018)**" mentioned by the reviewer is supplemented into Reference.

Comment #7:

The "visualization degree" seems to be a subjective measure of image quality. The use of more objective, quantitative metrics like PSNR is encouraged. CIFAR-10 accuracy might not be the best choice unless its relevance to the use-case scenario is explained.

Response:

We appreciated the Reviewer for the valuable suggestion. According to the Reviewer's suggestion, we added the calculated PSNR metrics to all the reconstructed images in the revised manuscript for objective evaluation.

We agree with the Reviewer's viewpoint that selecting previously unseen dataset for decompression and recognition is better. We replaced the Cifar-10 dataset with the ImageNet dataset since it contains richer images of daily life and a variety of image types unrelated to the training dataset (Cifar-10). Fig. R9 shows the comparisons

between original and reconstructed images randomly selected from the ImageNet dataset, in which we can find that sufficiently high PSNR values are achieved when the data is unseen. We adopted the ResNet34 neural network to recognize the original and reconstructed ImageNet datasets, as shown in Fig. R10. The decrease in recognition accuracy is slight (0.06%). We also explored the recognition accuracy of the ResNet34 neural networks that the testing and training datasets are not the same type, as shown in Fig. R11. The comparable recognition accuracies of unseen data demonstrate the practicability and feasibility of the proposed storage system.

Fig. R9 Comparison of original and reconstructed images randomly selected from the original and reconstructed ImageNet datasets.

Fig. R10 Recognition accuracy of the ResNet34 neural network trained and tested with the original and reconstructed ImageNet datasets.

Fig. R11 Recognition accuracy of the ResNet34 neural network trained with **a**, Original and **b**, Reconstructed ImageNet dataset, and tested with different types of datasets.

The following revisions have been made in the main text and supplementary information.

- 1) We supplemented the PSNR metrics in Supplementary Fig. 10.
- 2) The datasets used for image recognition application has been changed from Cifar-10 to ImageNet. The sample images in Supplementary Fig. 11 have been updated.
- 3) Page 15, line 430, lines 431-433. The deep neural network used for image recognition has been changed from ResNet18 to ResNet34 to obtain a reasonable accuracy for the ImageNet dataset. The recognition accuracy in Fig. 5a and Supplementary Fig. 13 are also updated accordingly.
- 4) Pages 12-13, lines 343-353. The evaluation results are updated and rewritten as: “The practicability and fidelity of the image data passing through the proposed near-storage in-memory processing system are vital metrics for data storage and subsequent practical applications. We deployed the ImageNet dataset and Kodak24 dataset, which contain richer images of daily life and various image types unrelated to the training dataset (Cifar-10), to examine the restoration degree of the trained storage system in the face of unseen images. Visualization results of the randomly selected sample images after compression are the same as those before compression (see Supplementary Fig. 11 and Fig. 12), and all the images obtain >33 dB PSNR values, which proves the universality of the proposed storage system and its ‘data class’-independent characteristic. Moreover, we used reconstructed and original ImageNet datasets to perform image recognition tasks with the ResNet34 neural network (see Methods)”.

Comment #8:

Fig. 3c: Please explain why PSNR decreases with more codec layers. Is it due to higher compression ratio leading to more information loss?

Response:

Yes, a higher compression ratio will lead to more information loss. So, the PSNR decreases with the increase of codec layers since the CAE network is a lossy compression technique. The information loss mainly results from the downsampling operation in convolution computing process. Although the primary feature information is extracted, some information will still be lost partly. As the number of layers increases, the lost information is accumulated, which leads to a decrease in the PSNR value after reconstruction. This issue can be alleviated by optimizing the network structure or training scheme, such as nonlinear transformation [20] or content-weighted scheme [21] so that a large-scale network can be used to obtain a high compression ratio while achieving an accepted PSNR value.

Corresponding explanation has been added in the manuscript as follows.

Page 8, lines 224-229. The explanations of why PSNR decreases with more codec layers is added: “It is worth noting that the PSNR value decreases with more codec layers, mainly due to partial information loss during downsampling operations in convolution computing process. This issue can be alleviated by optimizing the network structure or training scheme, such as nonlinear transformation [15] or content-weighted scheme [13] so that a large-scale network can be used to obtain a high compression ratio while achieving an accepted PSNR value”.

Overall Comment:

The overall concept and experimental demonstration of your work are very promising. I hope that addressing the above points will improve the manuscript and its potential impact on the scientific community.

Response:

We are deeply indebted to the Reviewer for providing valuable and constructive comments and suggestions. We have carefully addressed the above comments and conducted extensive data supplementation and discussions. The quality of the revised manuscript has been enhanced, and we hope it will showcase its potential impact on the scientific community.

Reference

- [1] Goyal, Y. et al. Making the v in vqa matter: Elevating the role of image understanding in visual question answering. in *Proceedings of the IEEE conference on computer vision and pattern recognition (CVPR)* (IEEE, 2017).
- [2] Intel Corporation. *Intel Power Gadget*, [Online]. Available: <https://software.intel.com/content/www/us/en/develop/artsoftware.intel.com/content/www/us/en/develop/articles/intel-power-gadget.html>.

- [3] Zheng, X. et al. Error-resilient analog image storage and compression with analog-valued RRAM arrays: an adaptive joint source-channel coding approach. in *2018 International Electron Devices Meeting (IEDM)* (IEEE, 2018).
- [4] SeaGate Technology. www.seagate.com/www-content/datasheets/pdfs/exos-7-e8-msft-data-sheet-DS1957-4M-1909US-en_GB.pdf.
- [5] Lv, H. et al. BEOL based RRAM with one extra-mask for low cost, highly reliable embedded application in 28 nm node and beyond. in *2017 IEEE International Electron Devices Meeting (IEDM)* (IEEE, 2017).
- [6] Shafiee, A. et al. ISAAC: A convolutional neural network accelerator with in-situ analog arithmetic in crossbars. *ACM Comput. Archit. News* **44**, 14-26 (2016).
- [7] Liu, Q. et al. A fully integrated analog ReRAM based 78.4 TOPS/W compute-in-memory chip with fully parallel MAC computing. in *2020 IEEE International Solid-State Circuits Conference (ISSCC)* (IEEE 2020).
- [8] Dong, X. et al. A circuit-architecture co-optimization framework for exploring nonvolatile memory hierarchies. *ACM T. Archit. Code Op.* **10**, 1-22 (2013).
- [9] Zangeneh, M. et al. Design and optimization of nonvolatile multibit 1T1R resistive RAM. *IEEE Trans. Very Large Scale Integr. VLSI Syst.* **22**, 1815-1828 (2013).
- [10] Muralimanohar, N. et al. CACTI 6.0: A tool to model large caches. *HP laboratories* **27**, 28 (2009).
- [11] Tripathi, V. et al. An 8-bit 450-MS/s single-bit/cycle SAR ADC in 65-nm CMOS. in *2013 Proceedings of the ESSCIRC (ESSCIRC)* (IEEE, 2013).
- [12] Kim, S. et al. A 65-nm CMOS 6-bit 20 GS/s time-interleaved DAC with full-binary sub-DACs. *IEEE T. Circuits-II* **65**, 1154-1158 (2018).
- [13] Rao, M. et al. Thousands of conductance levels in memristors integrated on CMOS. *Nature* **615**, 823-829 (2023).
- [14] Qi, C. et al. Low cost and highly reliable radiation hardened latch design in 65 nm CMOS technology. *Microelectron. Reliab.* **55**, 863-872 (2015).
- [15] Senthilpari, C. et al. Proposed low power, high speed adder-based 65-nm Square root circuit. *Microelectron. J.* **42**, 445-451 (2011).
- [16] Hardavellas, N. et al. Toward dark silicon in servers. *IEEE Micro* **31**, 6-15 (2011).
- [17] Jeong, B. H. et al. A 1.35 V 4.3 GB/s 1Gb LPDDR2 DRAM with controllable repeater and on-the-fly power-cut scheme for low-power and high-speed mobile application. in *2009 IEEE International Solid-State Circuits Conference (ISSCC)* (IEEE, 2009).
- [18] Li, C. et al. Analogue signal and image processing with large memristor crossbars. *Nat. Electron.* **1**, 52-59 (2018).
- [19] Zarccone, R. et al. Joint source-channel coding with neural networks for analog data compression and storage. in *2018 Data Compression Conference (DCC)* (IEEE,

2018).

- [20] Ballé, J. et al. End-to-end optimized image compression. in *2017 International Conference on Learning Representations (ICLR)* (2017).
- [21] Li, M. et al. Learning content-weighted deep image compression. *IEEE T. Pattern Anal.* **43**, 3446-3461 (2021).

Responses to the Reviewer 2

Overall Comment:

The authors present a storage system based on memristor which uses an autoencoder for compressing data. Both storage and the autoencoder stages are realized with memristors. The paper is well-written, and the topic is relatively new. Experiments are provided to demonstrate the concept in practice. However, some issues should be addressed before considering publication:

Response:

We are incredibly grateful to the Reviewer for the careful reading, positive comments, and constructive suggestions, which significantly improve the quality of our manuscript. We have carefully considered all the comments from the Reviewer and revised our manuscript accordingly. Below are the point-by-point responses to the Reviewer's comments and questions. We hope our responses have addressed all the concerns.

Comment #1:

The mapping of high precision weights is not clear in Supplementary Fig. 1. Why for example the weight $W_{\{1,1\}} = 4$ is converted to $[0010 - 0000]$ and not $[0100 - 0000]$ since the number 4 converted in binary with 4 bits is 0100, or $0100_2 = 4_{10}$?

Response:

We are sorry for the unclear illustration. In the previous manuscript, we used the 2-bit memristor for weight mapping. So, the mapping data is the quaternary digit. The decimal 4 can be represented as 0010 in quaternary code. Therefore, the decimal weight $W_{\{1,1\}} = 4$ is converted to $[+0010, -0000]$, where the first four digits represent positive weights and the last four digits represent negative weights.

In the revised manuscript, we replaced the 2-bit memristor with the 4-bit memristor for the experimental demonstration so that the bits used for weight mapping, data storage, and programming (Fig. 2c in the manuscript) are all the same. We have modified the schematic diagram of weight mapping scheme (shown in Fig. R1) and rewritten the corresponding illustration in the main text and supplementary information for a better understanding as follows.

Fig. R1 Weight mapping scheme. **a**, Mapping of signed 8-bit kernel weights into 4-bit memristor arrays. The 4-bit memristor has sixteen conductance states and thus can represent one hexadecimal digit. In this case, we can use two 4-bit memristors to represent one 8-bit weight. For example, decimal 4 is converted into hexadecimal as 04, representing $16^1 \times 0 + 16^0 \times 4 = 4$. Hence, each signed weight can map to four memristor columns: two columns for positive weights with positive coefficients on output and the other for negative weights with negative coefficients on output. **b**, Typical kernel weights mapping process. After training in the software, the 32-bit floating-point weights are quantized to signed 8-bit. Next, the weights are converted to two hexadecimal numbers. Considering the sign of the weights, if the weight is positive, it will be mapped to the positive two columns, and the negative two columns remain zero (ultra-low conductance). By precise programming, the weights in software are transferred into memristor arrays.

The following revisions have been made in the manuscript.

The schematic diagram of weight mapping scheme in Supplementary Fig. 2 has been modified based on 4-bit memristor. More details are supplemented into the caption to make weight mapping process clearer.

Comment #2:

The precision of VMM from the Supplementary Fig. 4 is not clear. I suggest performing

a correlation plot of the SW (ideal) VMM and the measured experimental VMM.

Response:

Many thanks to the Reviewer for the constructive suggestion. We have supplemented the correlation plot of the software vector-matrix multiplication (VMM) and the measured experimental VMM to the revised manuscript to verify the precision of VMM. The results are shown in Fig. R2. The small ratio of RMSE to conductance range indicates the high precision of the VMM computations.

Fig. R2 Measured experimental VMM results of (a) G_x operator and (b) G_y operator with respect to ideal results.

The following revisions have been made in the manuscript.

- 1) Correlation plots of ideal VMM and experimental VMM are added in supplementary information as Supplementary Fig. 6b and 6c, respectively. The caption is also added accordingly.
- 2) Page 6, lines 179-180. The content to correlation plots is added to the manuscript as: “The high precision of the computation results in correlation plots of ideal VMM and experimental VMM indicates the effectiveness of the schemes”.

Comment #3:

Line 212, where do the ‘magic’ numbers of 8 bits for the weights and 6 bits for the data come from? Did the authors perform a design space exploration with different precisions?

Response:

We are sorry for the missed explanation about choices for quantization bit number of weights and compressed data. As the Reviewer said, we selected the quantization bit number according to the design space exploration with different precisions, which can be summarized as below.

In order to clearly clarify the reasons for the selection of weights and compressed data

precision, we supplemented the design space exploration data with different precisions in different convolutional autoencoder (CAE) network structures, as shown in Fig. R3. After training the CAE network with different structures in software, the weights and data after compression are directly quantized with different precisions. Peak signal-to-noise ratio (PSNR) value is used as a criterion to indicate the quality of the quantized image. It can be seen from the results that the PSNR value increases significantly as the weight quantization accuracy increases and tends to be saturated from 8-bit to 9-bit. Similarly, as the compressed data accuracy increases, the PSNR value firstly increases from 4-bit to 6-bit and then tends to be saturated from 6-bit to 7-bit. Considering the area overhead of the memristor-based storage array, we choose 8-bit precision for weight and 6-bit precision for compressed data for subsequent simulation optimization and hardware demonstration.

Fig. R3 Design space exploration with different weight and compressed data precisions. CAE networks with two-, four-, and six-layer are used. During evaluation, the weight precision changes from 5-bit to 9-bit and the compressed data varies from 4-bit to 7-bit. 32 dB is set as the lower limit at which the human eye has difficulty distinguishing between original and reconstructed images. L2-C4b corresponds to a system with two-layer CAE network and 4-bit compressed data precision.

The following revisions have been made in the manuscript.

- 1) The design space exploration for choosing precisions of weight and compressed data are added into supplementary information as Supplementary Fig. 8. The discussion is added into the caption.
- 2) Page 8, lines 206-208. We added the description of selecting weight and compressed data precisions based on design space exploration. The design space exploration is performed with different quantization precisions and CAE network structures for choosing applicable precisions of weight and compressed data (see Supplementary

Fig. 8).

Comment #4:

Line 218, Fig. 3b does not show performance improvement after retraining. In general, references to figure 3 seem odd.

Response:

We thank the Reviewer for pointing out this issue. We have carefully examined all references of the figures used in the manuscript.

The following revisions have been made in the manuscript.

- 1) Page 8, line 215. The reference to Fig. 3b at the end of sentence: "... after retraining is very limited" is deleted.
- 2) Page 9, line 235. To make the correspondence between the text and the figure more accurate, the reference to Fig. 3d is moved to the end of sentence "... around the target conductance level (Fig. 3d)".

Comment #5:

Line 233, is the digital readout proposed the same proposed in "Zidan, M.A., Jeong, Y., Lee, J. et al. A general memristor-based partial differential equation solver. Nat Electron 1, 411–420 (2018). <https://doi.org/10.1038/s41928-018-0100-6>; (see SI Fig 6)? And if not, what are the differences?

Response:

We apologize for the unclear statement about the digital readout. As the Reviewer said, the digital readout scheme in this study is the same as that proposed in "Nat Electron 1, 411–420 (2018)" [1] in principle. So, the digital readout scheme is not an innovative design. Our main contribution is to find out that the digital readout scheme can achieve high PSNR values compared with the widely adopted analogue readout scheme in various compression techniques, including JPEG and deep learning-based compression networks [2-4]. This is because the digital readout scheme can mitigate the deviations caused by the device and circuit before decompression.

We have rewritten the corresponding statements and added the reference [Nat Electron 1, 411–420 (2018)] in the revised manuscript for a better understanding, as listed below.

- 1) Page 7, lines 198-200. To make the description more accurate, the sentence "we propose effective ... decompression processes" is rewritten as "we propose effective strategies for algorithm-device codesign of decompression processes and introduce the digital readout scheme [39] into the CAE network".
- 2) Page 9, lines 237-240. We highlighted and added the reasons and advantages of

using the digital readout scheme in the manuscript, and cited relevant reference. We added sentence: “To eliminate this deviation, we utilized a digital readout method, which appears common for multi-bit memory systems, for hardware demonstration of deep learning-based image compression technique to improve the quality of reconstructed images”.

- 3) Page 26, lines 787-788. The reference “Zidan, M.A., Jeong, Y., Lee, J. et al. A general memristor-based partial differential equation solver. *Nat. Electron.* **1**, 411–420 (2018)” mentioned by the Reviewer is supplemented into Reference.

Comment #6:

Generally, it is not clear if the storage system was tested with a class previously unseen by the autoencoder. What happens if something new is seen? Is it able to reconstruct it or a more complicated network is needed? I think this is critical, a storage system should be independent of the ‘class’ of the data that is being stored.

Response:

We totally agree with the Reviewer’s viewpoint that a storage system should be independent of the ‘class’ of data that is being stored. Therefore, as suggested by the Reviewer, we used the storage system trained with the Cifar-10 dataset to compress the ImageNet dataset. Fig. R4 shows the comparison images randomly selected from the original and reconstructed ImageNet datasets, in which we can find that sufficiently high PSNR values are achieved when the data is previously unseen. We adopted the ResNet34 neural network to recognize the original and reconstructed ImageNet datasets to test the quality of reconstructed data, as shown in Fig. R5. The decrease in recognition accuracy is slight (0.06%). We also explored the recognition accuracy of the ResNet34 neural networks that the testing and training datasets are not the same type, as shown in Fig. R6. The comparable recognition accuracies of unseen data demonstrate the practicability and feasibility of the proposed storage system.

Besides, we also adopted the proposed storage system trained with the Cifar-10 dataset to compress and reconstruct the Kodak24 dataset, a typical dataset used in image compression. As shown in Fig. R7, all the images obtain acceptable PSNR values, proving the universality of the proposed system and its ‘data class’-independent characteristic.

Fig. R4 Comparison of original and reconstructed images randomly selected from the original and reconstructed ImageNet datasets.

Fig. R5 Recognition accuracy of the ResNet34 neural network trained and tested with the original and reconstructed ImageNet datasets.

Fig. R6 Recognition accuracy of the ResNet34 neural network trained with **a**, Original and **b**, Reconstructed ImageNet dataset, and tested with different types of datasets.

Fig. R7 Original and reconstructed Kodak24 dataset with PSNR values. All the images from kodim01 to kodim24 are compressed and decompressed by using the two-layer CAE network integrated storage system with optimized weight and compressed data precisions. Each image is in RGB format with 768×512 resolution.

The following revisions have been made in the manuscript.

- 1) The sample images of the reconstructed ImageNet dataset and its counterparts in original dataset are updated in supplementary information as Supplementary Fig. 11.
- 2) The PSNR values of Kodak24 dataset tested with the trained storage system is added into supplementary information as Supplementary Fig. 12.
- 3) Page 15, line 430. The reconstructed ImageNet dataset is further used to image recognition task. The data in Fig. 5a and Supplementary Fig. 13 as well as corresponding captions are updated.

- 4) Pages 12-13, lines 343-353. The evaluation results are updated and rewritten as: “The practicability and fidelity of the image data passing through the proposed near-storage in-memory processing system are vital metrics for data storage and subsequent practical applications. We deployed the ImageNet dataset and Kodak24 dataset, which contain richer images of daily life and various image types unrelated to the training dataset (Cifar-10), to examine the restoration degree of the trained storage system in the face of unseen images. Visualization results of the randomly selected sample images after compression are the same as those before compression (see Supplementary Fig. 11 and Fig. 12), and all the images obtain >33 dB PSNR values, which proves the universality of the proposed storage system and its ‘data class’-independent characteristic. Moreover, we used reconstructed and original ImageNet datasets to perform image recognition tasks with the ResNet34 neural network (see Methods)”.

Comment #7:

Line 362, a simulation of what would happen using memristors with more conductance states should be done to motivate the statement.

Response:

We appreciate the Reviewer’s suggestion. Yes, it would be more understandable if we could simulate the proposed storage system when using the memristor with more conductance states. Fig. R8 shows the simulated storage density improvement and PSNR value with different memristor conductance state numbers. The results imply that storage density increases with the conductance state numbers while the PSNR value of the reconstructed images remains constant. Below is the detailed analysis and discussion.

We discuss the storage density based on the area overhead used for data processing and storage of different systems. Detailed area evaluation is shown in the following **Evaluation method**. Fig. R8 is the simulated results, showing increased storage density with the increased conductance states. That is, more conductance states mean fewer memristors required for storing the same amount of data, which is conducive to improving storage density. In addition, it can be observed that when the memristor conductance state varies from 3-bit to 4-bit or 6-bit to 8-bit, the storage density improvement is almost unchanged. This is because each compressed 6-bit data is designated to be stored in an integer number of cells rather than storing multiple compressed data information within one cell, which could minimize the complexity of peripheral circuitry used for readout data processing. Meanwhile, the PSNR values remain unchanged, which are insusceptible to the variation of the storage capacity of the memristor. The results demonstrate that the memristor arrays with more

conductance states can improve the storage density without concerning about performance degradation.

Fig. R8 Relationships of **a**, Storage density improvement and **b**, PSNR value with respect to different memristor conductance state numbers.

Evaluation method

For area evaluation, we considered the area of memristor array not only for data storage but also for all encoding / decoding processing as well as corresponding peripheral circuitry in cores and banks. The storage density improvement is defined as the ratio of the total area overhead used for compression / retrieval and data storage. VisualQA v2.0 dataset [5] is used for evaluation. In the central processing unit (CPU)-based processing system, the original VisualQA v2.0 dataset requires 25 GB storage space for data storage, which needs the hard disk drive (HDD) disk and its peripheral circuit area of about 7194 mm² for storage [6]. All the circuit components in systems are evaluated under 65 nm technology node. In this case, the areas of CPU and dynamic random-access memory (DRAM) for processing are estimated to be 2.48 and 84.23 mm² [7,8]. The architecture of the proposed near-storage in-memory processing system is shown in Fig. R9. The system consists of 8 in-memory computing banks, and the number of storage banks depends on the storage capacity of memristors. Each computing or storage bank consists of 16 cores. The size of the memristor array is set to be 1-Mb (1024×1024) [9] in the storage core, and that in computing core remains 4-kb (256×16) as demonstrated in the manuscript.

Fig. R9 Diagram of memristor-based near-storage in-memory processing architecture. **a**, Computing bank and a memristor-based computing core. **b**, Storage bank and internal structure of a storage core.

In the memristor-based computing bank, the area of buffer is evaluated to be 1.36 mm^2 by using CACTI [10] with 64-KB size and 2-kb bandwidth allowing for parallel computing of all computing cores. The digital-to-analogue converter (DAC) in [11] is adopted, which reported an active area of 0.072 mm^2 . In the computing core, register is designed based on D flip flop (DFF) [12], which has 8-bit for each of the word line (WL). The performance of MUX and sample and hold (S & H) circuits are simulated and optimized in Cadence. For analogue-to-digital converter (ADC), we used the data from [13]. The shift & add circuit is designed to realize 16-bit shift addition according to the adder and DFF acquired in references [12, 14]. Based on the above considerations, the area metrics of a computing core are summarized in Table R1. In memristor-based storage core, the area of one one-transistor one-memristor (1T1M) cell is set to $12F^2$ during evaluation, where F represents the feature size. In this case, the area of WL, bit line (BL), and source line (SL) drivers and MUX, which are simply composed of inverters or gate circuits, can be estimated based on the feature size, as shown in Table R2. CAE networks with two-, four-, and six-layers (see Methods) are used for evaluation. The memristors are capable of 2-bit, 3-bit, 4-bit, 6-bit, and 8-bit during evaluation to motivate the statement in the manuscript.

Table R1. Area metrics of each circuit components in one computing core with 4-kb (256×16) memristor array.

Module	Area (μm^2)
Register	22856
WL-MUX	21504
Array	9646
SL/BL-MUX	6208
S & H	2080
ADC	560000

Shift & adder	185600
---------------	--------

Table R2. Area metrics of each circuit components in one storage core with 1-Mb (1024×1024) memristor array.

Module	Area (μm^2)
WL driver	155.6
Array	53162
SL/BL driver	413.6
SL/BL-MUX	2998.6

The following revisions have been made in the manuscript.

- 1) The simulation results are added to supplementary information as Supplementary Fig. 14 to motivate the statement. Corresponding discussions are also supplemented into the figure caption.
- 2) Pages 13, lines 372-376. The sentence “However, higher storage density requires an encoder network with more compression layers to increase the compression ratio and memristor arrays with more conductance states” is rewritten as “It can be inferred from these results that by using deep neural networks with higher compression ratios or memristor arrays with more states, the storage density can be significantly improved (see Supplementary Fig. 14). However, the ensuing challenges should also be given due consideration” to enhance the overall coherence of the context.
- 3) Page 22-23, lines 661-690. The **Evaluation method** is added into Methods-Storage density evaluation.

Comment #8:

Compared with ref 43 and 36 it seems that the novelty of this work is an experimental verification of the concept. While the effort is significant and important, could the authors clarify the main conceptual novelty (not technical, such as autoencoder vs convolutional autoencoder) of this work?

Response:

The Reviewer’s comment really means a lot to us. Besides the experimental verification of the concept, the main conceptual novelty of this work is the near-storage in-memory computing. Since the data storage, compression, and reconstruction are all performed by memristor arrays, the in-memory computing banks for encoder / decoder can be closely integrated with the storage bank. Thus, the latency and energy consumption can be improved by more than 20× and 180× compared with the server-grade CPU-based processing system, respectively, and 5.6× and 30× compared with the memristor-based

non-near-storage in-memory processing system, respectively, as shown in Fig. R11. Detailed energy consumption and latency evaluations are seen in the following **Evaluation method**. As suggested by the Reviewer, we have emphasized the conceptual novelty of near-storage in-memory computing and added the latency and energy consumption comparisons as well as the corresponding evaluation methods in the revised manuscript.

Fig. R11 **a**, Latency and **b**, Energy comparison of server-grade CPU-based processing system, non-near-storage in-memory processing system, and near-storage in-memory processing systems.

Evaluation method

The schematic diagrams of the system architectures used for evaluation are shown in Fig. R12. We still use the VisualQA v2.0 dataset [5] to evaluate the latency and energy. The evaluation starts from the input of the data for compression and ends with the output of the reconstructed data. Since compression techniques usually contain various domain transformations and complex lossless coding schemes (such as Huffman coding), in most cases, the compression is implemented based on CPU. In the CPU-based processing system, the data is input into the CPU, and perform the compression computation. The compressed data is fed into the hard disk drive (HDD) via the dynamic random-access memory (DRAM). For data retrieval, the stored data is fetched by the CPU via DRAM and output after the decode computation. In the memristor-based non-near-storage in-memory processing system, the data is input into the memristor-based in-memory computing bank to perform the data compression with the CAE algorithm and then stored into the HDD. The stored data is read from the HDD and fed into the in-memory computing unit for data retrieval. In our proposed system, the memristor-based in-memory computing bank compresses the input data and stores it into the memristor-based storage bank. When the data is required for retrieval, the data is read out from the memristor-based storage bank and fed into the memristor-based in-memory computing bank for decoding with the CAE algorithm. The energy consumption is the sum of the energy consumption of all components. Due to the dataset being processed in a pipeline fashion during compression / retrieval, the system's latency depends on the process that consumes the most data processing and

storage time.

Fig. R12 Schematic diagram of **a**, CPU-based processing system, **b**, Memristor-based non-near-storage in-memory processing system, and **c**, Memristor-based near-storage in-memory processing system architectures.

Server-grade CPU-based processing system:

We configured a Dell PowerEdge R750 rack server with two 28-core Intel Xeon 6330 (10nm) and sixteen 64 GB DDR4 SDRAM for latency and energy consumption evaluation. Intel® Power Gadget 3.6 software [15] is used to measure the latency and power / energy of the encoding and decoding processes. The system latency is obtained by measuring the actual running time of the program. The energy consumption is calculated as follows: thermal design power \times CPU usage of executing process \times system latency. The network structure of AE refers to [4], which has 4096 neurons in the input and output layers and 448 neurons in the hinder layer (compressed layer). The CAE network uses the two-layer structure demonstrated in the manuscript, which has 8 convolution kernels with size $3 \times 3 \times 3$ and 3 transpose convolution kernels with size $2 \times 2 \times 8$. Both AE and CAE networks are performed in the Pytorch framework. Since the data storage and evaluation are performed on Dell PowerEdge R750 with SEAGATE (ST4000NM00A) HDD as storage media, the parameters from the SEAGATE datasheet [6] are used for calculation. The latency and energy consumption generated by the data access, writing / reading are estimated according to the interface access speed, max sustainable transfer rate, and average operating power, which can be acquired from the datasheet of 1.5 Gb/s, 215 MB/s, and 10.77 W, respectively. Based on the above considerations and parameters, the evaluation results on latency and energy consumption of server-grade CPU-based processing system are listed in Table R3 and Table R4, respectively.

Table R3. Latency of server-grade CPU-based processing system.

Compression technique	Process	CPU + DRAM (s)	Storage (s)
JPEG	Encode	535.98	42.12
	Decode	213.94	42.12
PNG	Encode	2072.07	240.33
	Decode	344.28	240.33
AE	Encode	1066.87	27.29
	Decode	1028.09	27.29
CAE	Encode	2267.26	124.8
	Decode	1061.15	124.8

Table R4. Energy consumption of server-grade CPU-based processing system.

Compression technique	Process	CPU + DRAM (J)	Storage (J)
JPEG	Encode	3884.74	212.99
	Decode	1550.71	212.99
PNG	Encode	15171.14	1210.6
	Decode	2520.63	1210.6
AE	Encode	43491.58	136.87
	Decode	41910.66	136.87
CAE	Encode	92808.36	626
	Decode	43437.22	626

Memristor-based near-storage in-memory processing system:

The architecture and configuration structure of memristor-based computing and storage bank are the same as those in comment #7. Memristor with 4-bit storage capacity is used during evaluation, as demonstrate in Fig. 2c in the main text. The latency and energy consumption are dominated by memristor array, ADC, DAC, and buffer, which are mainly considered during evaluation [16-19]. During the computing process, the weights of the two-layer CAE network can be completely mapped into one computing core, so one bank can map 16 sets of weights and can realize parallel computing. By utilizing two-layer CAE network, the VisualQA v2.0 dataset is compressed to 16.67×10^9 pixels with 6-bit accuracy for each pixel. In this case, the compressed data requires about 2133 storage banks for data storage, and different storage banks can

perform parallel writing or reading of data. Accordingly, 8 computing banks are integrated during evaluation to boost speed.

All the circuit components in the near-storage in-memory processing system are evaluated under 65 nm technology node. A 0.2 V read voltage with a 20 ns pulse width is adopted for the single-bit calculation of the memristor array. The mean weight value, extracted from the conductance distribution of all cells in the weight matrix, represents the weight of each cell is used for calculating the energy consumption. Similarly, the conductance value used for data writing and reading is extracted from the conductance distribution of the compressed data matrix. For data writing, the pulse numbers used for multi-bit memristor programming are calculated according to the total programming pulses (Fig. R13). We use CACTI [10] to evaluate the latency and energy consumption of the buffer, which is designed to be 64 KB in size and 2-kb in bandwidth to allow for parallel computing of all 16 cores in the computing bank. The ADC in [13] is adopted, which reported an 8-bit resolution with a sampling rate of 450 MS/s and an active area of 0.035 mm². 16 source lines (SLs) are operating in parallel with one ADC on each. We use the DAC data from [11], which reported 136 mW for digital / analog conversion under 20 GS/s. All the parameters are summarized in Table R5. According to these metrics, the latency and energy consumption for compressing and decompressing the VisualQA v2.0 dataset are calculated in Table R6.

Fig. R13 Statistical pulse numbers to program 4-bit memristor. Programming pulses of 100 cells are counted for each conductance state.

Table R5. Summary of metrics of key circuit modules in memristor-based computing bank used for evaluation.

Module	Energy	Latency	Area (μm^2)
ADC	14.3 pJ/bit	2.2 ns/op	35000
DAC	6.8 pJ/conversion	50 ps/op	72000

Buffer	0.38 pJ/bit	1.04 ns/access	1360000
Array	9.2 fJ/cell (encoding) 8.2 fJ/cell (decoding) 17 fJ/cell (reading) 25 pJ/cell (writing)	20 ns/pulse @ reading 100 ns/pulse @ writing	9646 @ 4-kb

Table R6. Evaluation results of latency and energy during data compression and retrieval with memristor-based near-storage in-memory processing system.

Performance	Process	Buffer	Array	ADC	Data write/access	Overall
Latency (s)	Encoding	0.27	10.41	0.57	22.14	22.14
	Decoding	0.3	11.71	0.64	0.67	11.71
Energy (J)	Encoding	1.37	0.13	7.63	11.66	20.79
	Decoding	0.46	0.04	8.58	0.7	9.78

Memristor-based non-near-storage in-memory processing system:

Since the non-near-storage in-memory processing approach combines memristor-based in-memory computing bank and HDD storage, the latency and energy consumption can be calculated based on available device and circuit metrics setting in the above discussion.

The following revisions have been made in the manuscript.

- 1) Page 15, line 430, lines 439-443. The evaluation results on latency and energy are added into manuscript as Fig. 5d and 5e. Corresponding captions are also supplemented.
- 2) Page 20-22, lines 581-660. The **Evaluation method** is added into Methods-System overhead evaluation.
- 3) Pages 14, lines 395-412. The evaluation results are added into manuscript, which is written as: “We compared the latency and energy consumption of the proposed system with that of other competitive technologies, including JPEG, PNG, autoencoder (AE), and CAE implemented in the server-grade CPU-based processing system, and CAE implemented in the non-near-storage in-memory processing system (see Supplementary Fig. 16 and Methods). Since compression techniques usually contain various domain transformations and complex lossless coding schemes (such as Huffman coding), in most cases, the compression is implemented based on the CPU. As illustrated in Fig. 5d and 5e, the near-storage in-memory processing system effectively reduces the latency and energy consumption by 20× and 180× compared with server-grade CPU-based processing

systems, respectively, and 5.6× and 30× compared with non-near-storage in-memory processing system, respectively. The significant improvements in energy efficiency can be mainly attributed to the efficient memristor-based in-memory computing technique and reduction in data transfer between computing unit and storage, manifesting the proposed system's considerable potential and energy efficiency advantages in data compression and retrieval. In addition, it can also be inferred that the latency of the compression/reconstruction process is primarily consumed by both data computing and storage. While the storage process mainly dominates energy consumption”.

- 4) The following statements are added or modified to emphasize the near-storage in-memory computing.

Page 1, lines 13-16. “Here, we propose a memristor-based storage system with an integrated near-storage in-memory computing-based convolutional autoencoder compression network to boost the energy efficiency and speed of the image compression / retrieval and improve the storage density”.

Page 3, lines 66-68. “This study proposes a memristor-based storage system integrated with the near-storage in-memory computing for data compression / decompression and multi-bit storage banks”.

Page 16, lines 447-450. “In this study, we proposed a memristor-based storage system that integrates a deep image compression network for image data storage, which is implemented based on a near-storage in-memory processing architecture with high efficiency and storage density”.

Minor comments:

Comment #1:

Line 168, the sentence is a bit confusing, it reads better as “a 4kb (256x16) crossbar arrays with TiN/TaO_x/HfO_x/TiN structure is fabricated on top of the CMOS circuits”.

Response:

Thanks for the Reviewer's careful reading. According to the Reviewer's suggestion, on page 6, lines 160-162, we rewritten the sentence “A 4-kb (256×16) crossbar array is fabricated on top of the CMOS circuits with TiN/TaO_x/HfO_x/TiN structure” to “**A 4-kb (256×16) crossbar array with TiN/TaO_x/HfO_x/TiN structure is fabricated on top of the CMOS circuits**”. Besides, we tried our best to improve the readability of the whole manuscript. These changes will not influence the content and structure of the manuscript. We did not list the changes here but marked them in the highlighted version.

Reference

- [1] Zidan, M. et al. A general memristor-based partial differential equation solver. *Nat. Electron.* **1**, 411-420 (2018).
- [2] Li, C. et al. Analogue signal and image processing with large memristor crossbars. *Nat. Electron.* **1**, 52-59 (2018).
- [3] Zarcone, R. et al. Joint source-channel coding with neural networks for analog data compression and storage. in *2018 Data Compression Conference (DCC)* (IEEE, 2018).
- [4] Zheng, X. et al. Error-resilient analog image storage and compression with analog-valued RRAM arrays: an adaptive joint source-channel coding approach. in *2018 International Electron Devices Meeting (IEDM)* (IEEE, 2018).
- [5] Goyal, Y. et al. Making the v in vqa matter: Elevating the role of image understanding in visual question answering. in *Proceedings of the IEEE conference on computer vision and pattern recognition (CVPR)* (IEEE, 2017).
- [6] SeaGate Technology. www.seagate.com/www-content/datasheets/pdfs/exos-7-e8-msft-data-sheet-DS1957-4M-1909US-en_GB.pdf.
- [7] Hardavellas, N. et al. Toward dark silicon in servers. *IEEE Micro* **31**, 6-15 (2011).
- [8] Jeong, B. H. et al. A 1.35 V 4.3 GB/s 1Gb LPDDR2 DRAM with controllable repeater and on-the-fly power-cut scheme for low-power and high-speed mobile application. in *2009 IEEE International Solid-State Circuits Conference (ISSCC)* (IEEE, 2009).
- [9] Lv, H. et al. BEOL based RRAM with one extra-mask for low cost, highly reliable embedded application in 28 nm node and beyond. in *2017 IEEE International Electron Devices Meeting (IEDM)* (IEEE, 2017).
- [10] Muralimanohar, N. et al. CACTI 6.0: A tool to model large caches. *HP laboratories* **27**, 28 (2009).
- [11] Kim, S. et al. A 65-nm CMOS 6-bit 20 GS/s time-interleaved DAC with full-binary sub-DACs. *IEEE T. Circuits-II* **65**, 1154-1158 (2018).
- [12] Qi, C. et al. Low cost and highly reliable radiation hardened latch design in 65 nm CMOS technology. *Microelectron. Reliab.* **55**, 863-872 (2015).
- [13] Tripathi, V. et al. An 8-bit 450-MS/s single-bit/cycle SAR ADC in 65-nm CMOS. in *2013 Proceedings of the ESSCIRC (ESSCIRC)* (IEEE, 2013).
- [14] Senthilpari, C. et al. Proposed low power, high speed adder-based 65-nm Square root circuit. *Microelectron. J.* **42**, 445-451 (2011).
- [15] Intel Corporation. *Intel Power Gadget*, [Online]. Available: <https://software.intel.com/content/www/us/en/develop/artsoftware.intel.com/content/www/us/en/develop/articles/intel-power-gadget.html>.
- [16] Shafiee, A. et al. ISAAC: A convolutional neural network accelerator with in-situ

- analog arithmetic in crossbars. *ACM Comput. Archit. News* **44**, 14-26 (2016).
- [17] Liu, Q. et al. A fully integrated analog ReRAM based 78.4 TOPS/W compute-in-memory chip with fully parallel MAC computing. in *2020 IEEE International Solid-State Circuits Conference (ISSCC)* (IEEE 2020).
- [18] Dong, X. et al. A circuit-architecture co-optimization framework for exploring nonvolatile memory hierarchies. *ACM T. Archit. Code Op.* **10**, 1-22 (2013).
- [19] Zangeneh, M. et al. Design and optimization of nonvolatile multibit 1T1R resistive RAM. *IEEE Trans. Very Large Scale Integr. VLSI Syst.* **22**, 1815-1828 (2013).

Responses to the Reviewer 3

Overall Comment:

This work presents an experimental demonstration of image compression with multi-bit resistive memories with packaged chip measurements. The idea essentially combines two aspects to achieve high storage density of compressed data: convolutional autoencoder network as the compression technique and utilizing multi-level cell capability of metal-oxide resistive memories. This is a comprehensive study of the feasibility of multi-bit resistive RAM for competing with existing high-density data storage solutions. However, the reviewer found that the work is not sufficiently novel to justify publication in this journal at the current stage, as both aspects were well studied in the prior works and this paper fails to show a significant leap towards more possibilities or new applications. Specific comments are listed below that might help with improving the overall quality for submission to other venues.

Response:

We would appreciate the Reviewer's feedback and suggestions. We feel regretful that the Reviewer did not get the contributions of this work, which may be caused by our unclear statements. The main contributions of this work can be summarized below.

(1) To the best of our knowledge, we first demonstrate the fully memristor-based storage in which encoding (convolution computation), compressed data storage and access, and decoding (transpose convolution computation) are all hardware implemented with the 4-bit memristor arrays. Although previous publications [1-4] have also demonstrated the data compression or storage of compressed data using memristor arrays, the compression techniques are different, and only partial functions were demonstrated with memristor arrays. Besides, the achieved peak signal-to-noise ratio (PSNR) value, the critical metric for data compression, is the highest among the non-volatile memory (NVM)-based data compression systems. Table R1 (Table 1 in the main text) compares our work with previous hardware implementations of image compression based on emerging NVMs, in a which significant leap in the functions and PSNR values can be observed.

Table R1. Summary of hardware implementation of image compression based on emerging NVMs.

Image compression technique	Compression	Decompression	Storage	Reconstructed image quality
JPEG ^[1]	Memristor (for DCT)	Software	-	-
_[2]	Memristor (for DFT)	Memristor (for IDFT)	-	26.1 dB

Fully-connected AE ^[3]	Software	Software	PCM	25.1 dB
Fully-connected AE ^[4]	Software	Software	Memristor	30.3 dB
Convolutional AE (this work)	Memristor	Memristor	Memristor	34.08 dB

DCT: discrete cosine transformation; DFT: discrete fourier transformation; IDFT: inverse discrete fourier transformation PCM: phase-change memory; AE: autoencoder

(2) The proposed storage system is a near-storage in-memory computing system, which is a relatively novel concept. Since the data storage, compression, and reconstruction are all performed by memristor arrays, the in-memory computing banks for encoder / decoder can be closely integrated with the storage bank. Thus, the energy and time consumed by the data movement can be mitigated. Therefore, the latency and energy consumption of the proposed system are improved by more than $20\times$ and $180\times$ compared with the server-grade central processing unit (CPU)-based processing system, respectively, and $5.6\times$ and $30\times$ compared with the memristor-based non-near-storage in-memory processing system, respectively, as shown in Fig. R1. Detailed latency and energy consumption evaluations are seen in the following **Evaluation method**. More importantly, the achievement in this work may raise inspiration for the design of other similar systems.

Fig. R1 **a**, Latency and **b**, Energy comparison of server-grade CPU-based processing system, non-near-storage in-memory processing system, and near-storage in-memory processing systems.

These contributions could fuel the hope of creating a new path to revolutionize the storage system, which is very important for modern information society, especially in the big data era. We have rewritten some statements to emphasize our contributions. We hope it will meet with the Reviewer's approval.

We have also carefully considered the reviewer's other comments and revised the manuscript accordingly. Detailed responses to the reviewer's comments are provided below point-by-point. We hope our responses have addressed all the concerns.

Evaluation method

The schematic diagrams of the system architectures used for evaluation are shown in Fig. R2. Here, we use the VisualQA v2.0 dataset [5], which is 25 GB with 265016 images, to evaluate the latency and energy. The evaluation starts from the input of the data for compression and ends with the output of the reconstructed data. In the CPU-based processing system, the data is input into the CPU, and perform the compression computation. The compressed data is fed into the hard disk drive (HDD) via the dynamic random-access memory (DRAM). For data retrieval, the stored data is fetched by the CPU via DRAM and output after the decode computation. In the memristor-based non-near-storage in-memory processing system, the data is input into the memristor-based in-memory computing bank to perform the data compression with the CAE algorithm and then stored into the HDD. The stored data is read from the HDD and fed into the in-memory computing unit for data retrieval. In our proposed system, the memristor-based in-memory computing bank compresses the input data and stores it into the memristor-based storage bank. When the data is required for retrieval, the data is read out from the memristor-based storage bank and fed into the memristor-based in-memory computing bank for decoding with the CAE algorithm. The energy consumption is the sum of the energy consumption of all components. Due to the dataset being processed in a pipeline fashion during compression / retrieval, the system's latency depends on the process that consumes the most data processing and storage time.

Fig. R2 Schematic diagram of **a**, CPU-based processing system, **b**, Memristor-based non-near-storage in-memory processing system, and **c**, Memristor-based near-storage in-memory processing system architectures.

Server-grade CPU-based processing system:

We configured a Dell PowerEdge R750 rack server with two 28-core Intel Xeon 6330 (10nm) and sixteen 64 GB DDR4 SDRAM for latency and energy consumption

evaluation. Intel® Power Gadget 3.6 software [6] is used to measure the latency and power / energy of the encoding and decoding processes. The system latency is obtained by measuring the actual running time of the program. The energy consumption is calculated as follows: thermal design power \times CPU usage of executing process \times system latency. The network structure of AE refers to [4], which has 4096 neurons in the input and output layers and 448 neurons in the hinder layer (compressed layer). The CAE network uses the two-layer structure demonstrated in the manuscript, which has 8 convolution kernels with size $3 \times 3 \times 3$ and 3 transpose convolution kernels with size $2 \times 2 \times 8$. Both AE and CAE networks are performed in the Pytorch framework. Since the data storage and evaluation are performed on Dell PowerEdge R750 with SEAGATE (ST4000NM00A) HDD as storage media, the parameters from the SEAGATE datasheet [7] are used for calculation. The latency and energy consumption generated by the data access, writing / reading are estimated according to the interface access speed, max sustainable transfer rate, and average operating power, which can be acquired from the datasheet of 1.5 Gb/s, 215 MB/s, and 10.77 W, respectively. Based on the above considerations and parameters, the evaluation results on latency and energy consumption of server-grade CPU-based processing system are listed in Table R2 and Table R3, respectively.

Table R2. Latency of server-grade CPU-based processing system.

Compression technique	Process	CPU + DRAM (s)	Storage (s)
JPEG	Encode	535.98	42.12
	Decode	213.94	42.12
PNG	Encode	2072.07	240.33
	Decode	344.28	240.33
AE	Encode	1066.87	27.29
	Decode	1028.09	27.29
CAE	Encode	2267.26	124.8
	Decode	1061.15	124.8

Table R3. Energy consumption of server-grade CPU-based processing system.

Compression technique	Process	CPU + DRAM (J)	Storage (J)
JPEG	Encode	3884.74	212.99
	Decode	1550.71	212.99
PNG	Encode	15171.14	1210.6

	Decode	2520.63	1210.6
AE	Encode	43491.58	136.87
	Decode	41910.66	136.87
CAE	Encode	92808.36	626
	Decode	43437.22	626

Memristor-based near-storage in-memory processing system:

The architecture of the memristor-based near-storage in-memory processing system consists of 2133 storage banks and 8 in-memory computing banks for VisualQA v2.0 dataset compression and storage, as shown in Fig. R3. Each computing and storage bank include 16 cores. The size of the memristor array is set to be 1-Mb (1024×1024) [8] in the storage core and 4-kb (256×16) in the computing core, as discussed in the manuscript. The latency and energy consumption are dominated by memristor array, analogue-to-digital converter (ADC), digital-to-analogue converter (DAC), and buffer, which are mainly considered during evaluation [9-12]. During the compression process, the weights of the two-layer CAE network can be completely mapped into one computing core, so one bank can map 16 sets of weights and realize parallel computing. The dataset is compressed to 16.67×10^9 pixels with 6-bit accuracy for each pixel by the two-layer CAE network. In this case, the compressed data requires about 2133 storage banks for data storage, and different storage banks can perform parallel writing or reading of data. Accordingly, 8 computing banks are integrated during evaluation to boost speed.

Fig. R3 Diagram of memristor-based near-storage in-memory processing architecture. **a**, Computing bank and a memristor-based computing core. **b**, Storage bank and internal structure of a storage core.

All the circuit components in the near-storage in-memory processing system are evaluated under 65 nm technology node. A 0.2 V read voltage with a 20 ns pulse width is adopted for the single-bit calculation of the memristor array. The mean weight value, extracted from the conductance distribution of all cells in the weight matrix, represents

the weight of each cell is used for calculating the energy consumption. Similarly, the conductance value used for data writing and reading is extracted from the conductance distribution of the compressed data matrix. For data writing, the pulse numbers used for multi-bit memristor programming are calculated according to the total programming pulses (Fig. R4). We use CACTI [13] to evaluate the latency and energy consumption of the buffer, which is designed to be 64 KB in size and 2-kb in bandwidth to allow for parallel computing of all 16 cores in the computing bank. The ADC in [14] is adopted, which reported an 8-bit resolution with a sampling rate of 450 MS/s and an active area of 0.035 mm². 16 source lines (SLs) are operating in parallel with one ADC on each. We use the DAC data from [15], which reported 136 mW for digital / analog conversion under 20 GS/s. All the parameters are summarized in Table R4. According to these metrics, the latency and energy consumption for compressing and decompressing the VisualQA v2.0 dataset are calculated in Table R5.

Fig. R4 Statistical pulse numbers to program 4-bit memristors. Programming pulses of 100 cells are counted for each conductance state.

Table R4. Summary of metrics of key circuit modules in memristor-based computing bank used for evaluation.

Module	Energy	Latency	Area (μm^2)
ADC	14.3 pJ/bit	2.2 ns/op	35000
DAC	6.8 pJ/conversion	50 ps/op	72000
Buffer	0.38 pJ/bit	1.04 ns/access	1360000
Array	9.2 fJ/cell (encoding) 8.2 fJ/cell (decoding) 17 fJ/cell (reading) 25 pJ/cell (writing)	20 ns/pulse @ reading 100 ns/pulse @ writing	9646 @ 4-kb

Table R5. Evaluation results of latency and energy during data compression and retrieval with memristor-based near-storage in-memory processing system.

Performance	Process	Buffer	Array	ADC	Data write/access	Overall
Latency (s)	Encoding	0.27	10.41	0.57	22.14	22.14
	Decoding	0.3	11.71	0.64	0.67	11.71
Energy (J)	Encoding	1.37	0.13	7.63	11.66	20.79
	Decoding	0.46	0.04	8.58	0.7	9.78

Memristor-based non-near-storage in-memory processing system:

Since the non-near-storage in-memory processing approach combines memristor-based in-memory computing bank and HDD storage, the latency and energy consumption can be calculated based on available device and circuit metrics setting in the above discussion.

The following revisions have been made in the manuscript.

- 1) The following statements are added or modified to emphasize the near-storage in-memory computing and highlight the high-quality hardware demonstration of the proposed system.

Page 1, lines 13-16. “Here, we propose a memristor-based storage system with an integrated near-storage in-memory computing-based convolutional autoencoder compression network to boost the energy efficiency and speed of the image compression / retrieval and improve the storage density”.

Page 1, lines 24-27. “Benchmark comparison results show that the 4-bit memristor-based storage system could reduce the latency and energy consumption by over 20× and 180× compared to the server-grade central processing unit (CPU)-based system and improve the storage density by more than 3 times”.

Page 3, lines 66-68. “This study proposes a memristor-based storage system integrated with the near-storage in-memory computing for data compression / decompression and multi-bit storage banks”.

Page 3, lines 78-81. “The evaluation results indicate that the proposed system effectively reduces latency and energy consumption by more than 20× and 180× compared with the server-grade central processing unit (CPU)-based processing system and improves storage density by more than 3 times”.

Pages 14, lines 395-412. The evaluation results are added into manuscript, which is written as: “We compared the latency and energy consumption of the proposed system with that of other competitive technologies, including JPEG, PNG, autoencoder (AE), and CAE implemented in the server-grade CPU-based

processing system, and CAE implemented in the non-near-storage in-memory processing system (see Supplementary Fig. 16 and Methods). Since compression techniques usually contain various domain transformations and complex lossless coding schemes (such as Huffman coding), in most cases, the compression is implemented based on the CPU. As illustrated in Fig. 5d and 5e, the near-storage in-memory processing system effectively reduces the latency and energy consumption by 20× and 180× compared with server-grade CPU-based processing systems, respectively, and 5.6× and 30× compared with non-near-storage in-memory processing system, respectively. The significant improvements in energy efficiency can be mainly attributed to the efficient memristor-based in-memory computing technique and reduction in data transfer between computing unit and storage, manifesting the proposed system's considerable potential and energy efficiency advantages in data compression and retrieval. In addition, it can also be inferred that the latency of the compression/reconstruction process is primarily consumed by both data computing and storage. While the storage process mainly dominates energy consumption”.

Page 15, lines 422-427. “Here, we experimentally demonstrate the fully memristor-based storage system equipped with image compression / decompression networks for the first time. Compression (convolution computing), compressed data storage and access, and decompression (transpose convolution computing) are all demonstrated accompanied by achieving the high PSNR value (34.08 dB) of reconstructed images”.

Page 16, lines 447-450. “In this study, we proposed a memristor-based storage system that integrates a deep image compression network for image data storage, which is implemented based on a near-storage in-memory processing architecture with high efficiency and storage density”.

Page 16, lines 457-460. “The benchmark comparison results illustrate that the memristor-based near-storage in-memory processing system can achieve reductions of over 20× in latency and 180× in energy consumption compared to the server-grade CPU-based processing system”.

- 2) Page 16, line 444. In Table 1, the PSNR of the reconstructed image during experimental demonstration is updated.
- 3) Page 15, line 430, lines 439-443. The evaluation results on latency and energy are added into manuscript as Fig. 5d and 5e. Corresponding captions are also supplemented.
- 4) Page 20-22, lines 581-660. The **Evaluation method** is added into Methods-System overhead evaluation.

Comment #1:

It's a bit misleading to readers when the authors use the word "deep learning" in the title with a two-layer shallow network (CAE).

Response:

Thanks for the Reviewer's suggestion. We agree with the Reviewer that the term "deep learning" in the title is a bit exaggerated and might be misleading to the readers. Therefore, we modified the title from "Memristor-based high-density storage system with deep learning-based image compression network" to "Memristor-based storage system with convolutional autoencoder-based image compression network" to avoid confusion.

The following revisions have been made in the manuscript.

Page 1, lines 1-2. The title is changed to: "Memristor-based storage system with convolutional autoencoder-based image compression network". The title in supplementary information is modified accordingly.

Comment #2:

Multi-bit conductance readout and conversion to analog width-modulated pulses are not discussed clearly. Such conversion overhead might be high to generate accurate, analog pulses.

Response:

We apologize for the ambiguous discussion and schematic diagram (Fig. 3d in the previous manuscript) of the data readout and conversion process. This study used the bit-slicing technique to convert the multi-bit conductance readout into the input for decoding. During decoding, the stored data is firstly read out from the memristor array and then quantized to 6-bit by the ADC module in the storage bank. In this case, the data output from the storage bank is binary, which can be directly input to the decoder bitwise from most significant bit (MSB) to least significant bit (LSB) and does not need any other peripheral circuitry for data conversion operation. The schematic diagram of the digital data readout process has been modified from analogue pulses to a series of binary pulses with fixed width when using digital readout scheme, as shown in Fig. R5.

Fig. R5 Schematic diagram of the analogue and digital readout and conversion processes.

The following revisions have been made in the manuscript.

- 1) Page 9, lines 241-245. The sentence “and then input ... discrete target conductance level” is rewritten as “**and then input into the transpose convolution network bitwise from the most significant bit (MSB) to the least significant bit (LSB) to perform decompression computations.** In this case, the distributed conductance values can be accurately quantified to the discrete target conductance level **without needing other peripheral circuitry for data conversion operation**”.
- 2) Page 10, line 273. The schematic diagram of Fig. 3d has been updated.
- 3) Page 10, lines 280-283. The caption of Fig. 3d is rewritten as “**Schematic diagram of the analogue and digital readout and conversion processes. The conductance values by using digital readout are quantized through ADC, which is essentially in the form of binary and can be directly input to the decoder network**”.

Comment #3:

With all the overhead considered, it is necessary to benchmark the RRAM-based solution with conventional digital solutions to show how the multi-bit capability can boost overall efficiency.

Response:

Many thanks for the Reviewer’s comment. According to the Reviewer’s suggestion, based on different memristor multi-bit capabilities, we compared the memristor-based solution with conventional digital solution in terms of latency, energy consumption, and area. VisualQA v2.0 dataset is used for all evaluations. As shown in Fig. R6, the results demonstrate that when employing the memristor-based solution for data compression / retrieval, with an increase in memristor’s multi-bit capability (2-bit to 8-bit), the latency and energy consumption overhead decreases by factors from 9× to 24× and 77× to 283×, respectively. In addition, the storage density is improved from 2× (two-layer CAE network with 2-bit memristor) to 18× (six-layer CAE network with 8-bit memristor)

according to the area ratio before and after compression (Fig. R7). The results indicate that the more multi-bit characteristics the memristors possess, the higher the system efficiency. The detailed discussions are as followings.

Fig. R6 **a**, Latency and **b**, Energy comparison server-grade CPU-based processing system and memristor-based near-storage in-memory processing system. The data compression and retrieval processes of JPEG, AE, and CAE techniques are assessed by using server-grade CPU-based processing approach. For near-storage in-memory processing, the storage system with 2-, 4-, and 8-bit memristor multi-bit capability is discussed using CAE network for compression and decompression.

Fig. R7 Area comparison of server-grade CPU-based processing system and proposed near-storage in-memory processing system for storing VisualQA v2.0 dataset. Different memristor multi-bit capabilities and CAE network structures are used for evaluation.

We consider memristor-based solution is the proposed near-storage in-memory processing system shown in Fig. R2c in overall comment, and the conventional digital solution is the server-grade CPU-based processing system (Fig. R2a). Considering system overheads including latency, energy consumption, and area, we discussed how the multi-bit capability of memristors contributes to boosting system performance.

Latency and energy consumption:

We assessed latency and energy consumption of compression / retrieval processes of the proposed storage system with different memristor multi-bit capabilities, and compared with those of JPEG, AE, CAE compression techniques implemented in server-grade CPU-based processing system. All the circuit metrics and evaluation methods are the same as those demonstrated in overall comment section, which can refer to Fig. R2, Fig. R3, Table R2, Table R3, and Table R4. Based on these metrics, the comparison results are shown in Fig. R6, which illustrate that the memristor-based solution has lower latency and energy consumption which can be further reduced with higher memristor multi-bit capabilities.

Area:

We also estimated the area of different storage systems under different memristor storage capacities. The detailed **Evaluation method** is described below.

Firstly, it must be clear that the storage density improvement is defined as the ratio of the total area overhead used for compression / retrieval and data storage. We considered the area of memristor array not only for data storage but also for all encoding / decoding processing as well as corresponding peripheral circuitry in cores and banks during evaluation. We still use the VisualQA v2.0 dataset [5] to evaluate the area. The architecture of the proposed system is shown in Fig. R2c and Fig. R3. The system consists of 8 in-memory computing banks, and the number of storage banks depends on the storage capacity of memristors. The parameters of buffer, ADC, DAC, and memristor array in computing bank could refer to Table R4. In the computing core, register is designed based on D flip flop (DFF) [16], which has 8-bit for each of the word line (WL). The performance of MUX and sample and hold (S & H) circuits are simulated and optimized in Cadence. The shift & add circuit is designed to realize 16-bit shift addition based on the adder and DFF acquired in references [16, 17]. The area metrics of a computing core are shown in Table R6. In the storage core, the area of one one-transistor one-memristor (1T1M) cell is set to $12F^2$ during evaluation, where F represents the feature size. In this case, the area of WL, bit line (BL), and SL drivers and MUX, which are simply composed of inverters or gate circuits, can be estimated based on the feature size, as shown in Table R7. The peripheral circuits including DAC, S & H, ADC, and shift & adder are located in the storage bank outside the core. The original VisualQA v2.0 dataset requires 25 GB storage space for data storage, which

needs the HDD disk and its peripheral circuit area about 7194 mm² for storage [7]. The areas of CPU and DRAM for processing are estimated to be 2.48 and 84.23 mm² [18, 19]. CAE networks with two-, four-, and six-layers (see Methods) are used for evaluation. The memristors are capable of 2-bit, 4-bit, and 8-bit storage capacities. Based on these considerations, the impact of multi-bit capabilities of memristor on enhancing storage density is shown in Fig. R7.

In summary, based on the proposed near-storage in-memory processing system, more storage capacities of memristor could promote the latency and energy efficiencies, and also effectively improve the storage density of the system, showcasing the potential of multi-bit memristors for efficient processing and storing of various data in big data era.

Table R6. Area metrics of circuit components in one computing core with 4-kb (256×16) memristor array.

Module	Area (μm ²)
Register	22856
WL-MUX	21504
Array	9646
SL/BL-MUX	6208
S & H	2080
ADC	560000
Shift & adder	185600

Table R7. Area metrics of circuit components in one storage core with 1-Mb (1024×1024) memristor array.

Module	Area (μm ²)
WL driver	155.6
Array	53162
SL/BL driver	413.6
SL/BL-MUX	2998.6

The following revisions have been made in the manuscript.

- 1) The detailed discussion on how the memristor multi-bit capability can boost overall efficiency of the storage system is added to supplementary information as Supplementary Note IV. Discussion on memristor multi-bit capabilities boosting system performance.
- 2) The evaluations of latency, energy consumption, and storage density improvement under different memristor multi-bit capabilities are added into supplementary information as Supplementary Fig. 17 and Fig. 18.

- 3) Page 14, lines 412-416. To introduce the contents of discussion on memristor multi-bit capabilities boosting overall efficiency of system, we added sentence: “Moreover, the impact of memristor multi-bit capability on boosting system performance is further discussed (see Supplementary Note IV). The results illustrate that memristor devices with more storage capacities could effectively promote the overall efficiency of the system (see Supplementary Fig. 17 and 18)”.
- 4) Page 22-23, lines 661-690. The **Evaluation method** is added into Methods-Storage density evaluation.

Reference

- [1] Li, C. et al. Analogue signal and image processing with large memristor crossbars. *Nat. Electron.* **1**, 52-59 (2018).
- [2] Han, Z. et al. Implementation of discrete Fourier transform using RRAM arrays with quasi-analog mapping for high-fidelity medical image reconstruction. in *2021 International Electron Devices Meeting (IEDM)* (IEEE, 2021).
- [3] Zarcone, R. et al. Joint source-channel coding with neural networks for analog data compression and storage. in *2018 Data Compression Conference (DCC)* (IEEE, 2018).
- [4] Zheng, X. et al. Error-resilient analog image storage and compression with analog-valued RRAM arrays: an adaptive joint source-channel coding approach. in *2018 International Electron Devices Meeting (IEDM)* (IEEE, 2018).
- [5] Goyal, Y. et al. Making the v in vqa matter: Elevating the role of image understanding in visual question answering. in *Proceedings of the IEEE conference on computer vision and pattern recognition (CVPR)* (IEEE, 2017).
- [6] Intel Corporation. *Intel Power Gadget*, [Online]. Available: <https://software.intel.com/content/www/us/en/develop/artsoftware.intel.com/content/www/us/en/develop/articles/intel-power-gadget.html>.
- [7] SeaGate Technology. www.seagate.com/www-content/datasheets/pdfs/exos-7-e8-msft-data-sheet-DS1957-4M-1909US-en_GB.pdf.
- [8] Lv, H. et al. BEOL based RRAM with one extra-mask for low cost, highly reliable embedded application in 28 nm node and beyond. in *2017 IEEE International Electron Devices Meeting (IEDM)* (IEEE, 2017).
- [9] Shafiee, A. et al. ISAAC: A convolutional neural network accelerator with in-situ analog arithmetic in crossbars. *ACM Comput. Archit. News* **44**, 14-26 (2016).
- [10] Liu, Q. et al. A fully integrated analog ReRAM based 78.4 TOPS/W compute-in-memory chip with fully parallel MAC computing. in *2020 IEEE International Solid-State Circuits Conference (ISSCC)* (IEEE 2020).
- [11] Dong, X. et al. A circuit-architecture co-optimization framework for exploring

- nonvolatile memory hierarchies. *ACM T. Archit. Code Op.* **10**, 1-22 (2013).
- [12] Zangeneh, M. et al. Design and optimization of nonvolatile multibit 1T1R resistive RAM. *IEEE Trans. Very Large Scale Integr. VLSI Syst.* **22**, 1815-1828 (2013).
- [13] Muralimanohar, N. et al. CACTI 6.0: A tool to model large caches. *HP laboratories* **27**, 28 (2009).
- [14] Tripathi, V. et al. An 8-bit 450-MS/s single-bit/cycle SAR ADC in 65-nm CMOS. in *2013 Proceedings of the ESSCIRC (ESSCIRC)* (IEEE, 2013).
- [15] Kim, S. et al. A 65-nm CMOS 6-bit 20 GS/s time-interleaved DAC with full-binary sub-DACs. *IEEE T. Circuits-II* **65**, 1154-1158 (2018).
- [16] Qi, C. et al. Low cost and highly reliable radiation hardened latch design in 65 nm CMOS technology. *Microelectron. Reliab.* **55**, 863-872 (2015).
- [17] Senthilpari, C. et al. Proposed low power, high speed adder-based 65-nm Square root circuit. *Microelectron. J.* **42**, 445-451 (2011).
- [18] Hardavellas, N. et al. Toward dark silicon in servers. *IEEE Micro* **31**, 6-15 (2011).
- [19] Jeong, B. H. et al. A 1.35 V 4.3 GB/s 1Gb LPDDR2 DRAM with controllable repeater and on-the-fly power-cut scheme for low-power and high-speed mobile application. in *2009 IEEE International Solid-State Circuits Conference (ISSCC)* (IEEE, 2009).

REVIEWER COMMENTS

Reviewer #1 (Remarks to the Author):

discussion on performance, rather than only reporting the compression ratio as in the initial version, and ensuring all discussions are based on a consistent bit-precision level. They have also incorporated several details to enhance the clarity of the manuscript.

However, the reviewer believes that neural network classification accuracy is not a necessary metric to evaluate the performance of image compression, and the use of DNN is somewhat excessive. The accuracy of DNNs does not inherently translate to an accurate reflection of human perception of image content. Moreover, if the sole aim of the compression is to enhance DNN classification accuracy, there exist more specialized algorithms that are specifically tailored for this purpose. The reviewer also appreciates the authors' efforts in evaluating the more complex ImageNet. Nevertheless, DNN accuracy remains a valid metric.

Given that all comments have been fully addressed, the reviewer is happy to support the publication of this work.

Reviewer #2 (Remarks to the Author):

The authors addressed all the concerns raised by this reviewer.

Reviewer #3 (Remarks to the Author):

The authors have made several improvements and clarifications. My previous comment #2 was not addressed. I was looking for an apple-to-apple comparison between RRAM-based near-storage, in-memory design (authors' work) vs. an equivalent digital ASIC design, yet what authors compared against was a generic CPU-based estimations that come with unnecessary overheads to support general-purpose tasks. Even if the authors were to pick a general-purpose platform for benchmarking in addition to digital ASIC baseline, I would argue that GPU platform would be a much better choice as the autoencoder would be easily handled by GPU cores and the memory capacity needed is relatively small.

The key rational behind asking for this apple-to-apple comparison, is to help authors illustrate why their "near-storage, in-memory" design can provide benefits compared to conventional "in-memory" RRAM accelerators. The reviewer would agree that making the in-memory component closer to the data storage itself is beneficial, but this benefit was not clearly quantitated in the current technical results.

Therefore, in order to truly prove this is a novel and efficient approach, more experiments would need to be provided in the benchmarking section.

RESPONSE TO REVIEWERS' COMMENTS

Response to the Reviewer 1

Overall Comment:

discussion on performance, rather than only reporting the compression ratio as in the initial version, and ensuring all discussions are based on a consistent bit-precision level. They have also incorporated several details to enhance the clarity of the manuscript.

However, the reviewer believes that neural network classification accuracy is not a necessary metric to evaluate the performance of image compression, and the use of DNN is somewhat excessive. The accuracy of DNNs does not inherently translate to an accurate reflection of human perception of image content. Moreover, if the sole aim of the compression is to enhance DNN classification accuracy, there exist more specialized algorithms that are specifically tailored for this purpose. The reviewer also appreciates the authors' efforts in evaluating the more complex ImageNet. Nevertheless, DNN accuracy remains a valid metric.

Given that all comments have been fully addressed, the reviewer is happy to support the publication of this work.

Response:

We sincerely thank the Reviewer's support and the valuable time the Reviewer has spent reviewing our manuscript and providing insightful comments to help significantly improve the quality of our work. We are happy that the Reviewer is satisfied with our revision.

Response to the Reviewer 2

Overall Comment:

The authors addressed all the concerns raised by this reviewer.

Response:

We highly appreciate the Reviewer for the valuable time in reviewing our manuscript and providing constructive comments, which significantly improve the quality of our work. We are delighted that the Reviewer is satisfied with our revision.

Responses to the Reviewer 3

Overall Comment:

The authors have made several improvements and clarifications. My previous comment #2 was not addressed. I was looking for an apple-to-apple comparison between RRAM-based near-storage, in-memory design (authors' work) vs. an equivalent digital ASIC design, yet what authors compared against was a generic CPU-based estimations that come with unnecessary overheads to support general-purpose tasks. Even if the authors were to pick a general-purpose platform for benchmarking in addition to digital ASIC baseline, I would argue that GPU platform would be a much better choice as the autoencoder would be easily handled by GPU cores and the memory capacity needed is relatively small.

The key rational behind asking for this apple-to-apple comparison, is to help authors illustrate why their "near-storage, in-memory" design can provide benefits compared to conventional "in-memory" RRAM accelerators. The reviewer would agree that making the in-memory component closer to the data storage itself is beneficial, but this benefit was not clearly quantitated in the current technical results.

Therefore, in order to truly prove this is a novel and efficient approach, more experiments would need to be provided in the benchmarking section.

Response:

We sincerely thank the Reviewer for the precious time and positive and constructive comments on our manuscript. We apologize for not fully addressing the Reviewer's previous comment #2. We have carefully studied it again and added explanations in the following response. In the benchmarking section, according to the Reviewer's suggestions, we supplement the performance evaluation results of systems based on the graphics processing unit (GPU) and application-specific integrated circuit (ASIC). Detailed responses to the Reviewer's comments are provided below. We hope our responses have addressed all the concerns.

Response to previous comment #2:

Previous comment #2: Multi-bit conductance readout and conversion to analog width-modulated pulses are not discussed clearly. Such conversion overhead might be high to generate accurate, analog pulses.

The conversion of multi-bit conductance readout from the memristor array into analogue width-modulated pulses can be designed as the architecture shown in Fig. R1. The current readout from the memristor array passes through a trans-impedance amplifier (TIA) to convert the current signal into the voltage signal. Then, the voltage-

to-time converter (VTC) is adopted to convert the analogue input voltage into an analogue pulse. The overhead of the circuit modules is evaluated under 65 nm CMOS technology. The TIA in [1] is adopted, which reported a 10 ns latency and 0.02 pJ energy consumption for each operation. We use the VTC module designed in [2], which shows a latency of 1.17 ns and estimated energy consumption of 0.96 pJ/op according to the formulas of the figure of merit (FOM) and the effective number of bits (ENOB) [3]. Due to the data processing being performed in a pipeline manner, the latency depends on the module that consumes the most. In this case, the latency and energy consumption of analogue readout scheme for one operation are 10 ns and 0.98 pJ, respectively. For the digital readout scheme, both latency and energy consumption during the readout process are dominated by the analog-to-digital converter (ADC). An ADC in [4] is used for evaluation, which exhibits a 2.2 ns on latency and 14.3 pJ/op on energy consumption. Based on the above discussion, the analogue readout scheme is more time-consuming, while the digital readout scheme consumes more energy.

Fig. R1 Architecture of memristor-based core for implementing analogue width-modulated pulse conversion.

The following revisions have been made in the manuscript.

- 1) The detailed discussion on analogue width-modulated pulses conversion process is added into supplementary information as Supplementary Note III. Conversion of analogue width-modulated pulse.
- 2) The architecture of memristor-based core used for implementing conversion of analogue width-modulated pulses are added into supplementary information as Supplementary Fig. 9.
- 3) Page 8, lines 237-238. The sentence on multi-bit conductance readout and conversion to analogue width-modulated pulse is added into manuscript, which is written as: “which is discussed in detail in Supplementary Note III and Fig. 9”.

Response to benchmarking comparison results:

Many thanks for the Reviewer's comments, which are very helpful in improving the convincingness of our results. According to the Reviewer's suggestion, we have evaluated the latency and energy consumption of the convolutional autoencoder (CAE) compression technique implemented with GPU-based and ASIC-based processing systems and compared it with those implemented with the near-storage in-memory processing system. We employed an NVIDIA Tesla T4 GPU and recently reported ASIC, CHIMERA accelerator [5], for evaluation. Fig. R2 shows the comparison results, which indicate that the proposed near-storage in-memory processing system reduces the latency and energy consumption by a factor of more than $5.6\times/5.6\times$ and $91\times/30\times$, respectively, compared with the GPU-based processing system/the ASIC-based processing system. It can be found that the latency of the GPU-based processing system is almost the same as that of the ASIC-based processing system since the latency of both processing systems is dominated by hard disk drive (HDD) storage. It can be analyzed from the results that integrating memristor-based computing banks and storage banks within the same unit effectively reduces the latency and energy consumption caused by circuit module interfaces and internal transfer during data movement. The evaluation results have been supplemented into the manuscript. The detailed **Evaluation method** is as follows.

Fig. R2 **a**, Latency and **b**, Energy comparison of GPU-based processing system, ASIC-based processing system, and near-storage in-memory processing system for implementing CAE compression technique.

Evaluation method

The schematic diagrams of the system architectures used for evaluation are shown in Fig. R3. Here, we use the VisualQA v2.0 dataset [6] for latency and energy consumption evaluations. The evaluation starts from the input of the data for compression and ends with the output of the reconstructed data. In the GPU (ASIC)-based processing systems, the data is input into the GPU (ASIC), to perform compression. The compressed data is stored into the HDD via the dynamic random-access memory (DRAM). For data retrieval, the stored data is fetched to the GPU (ASIC) through DRAM and output after decompression. In the proposed near-storage in-memory processing system, the

memristor-based in-memory computing bank compresses the input data and stores it into the memristor-based storage bank. When the data is required for retrieval, the data is read out from the memristor-based storage bank and fed into the memristor-based in-memory computing bank for decoding with the CAE algorithm. Due to the dataset being processed in a pipeline fashion during compression/retrieval, the system’s latency depends on the process that consumes the most data processing and storage time.

Fig. R3 Schematic diagram of **a**, GPU-based processing system, **b**, ASIC-based processing system, and **c**, Memristor-based near-storage in-memory processing system architectures.

GPU-based processing system:

We employed an NVIDIA Tesla T4 GPU (TSMC 12nm) backed with 16 GB GDDR6 DRAM to evaluate the latency and energy consumption. The system latency is obtained by measuring the actual running time of the program. The energy consumption is calculated as follows: (program execution power - system standby power) × system latency. The CUDA Toolkit [7] is used to perform CAE models on the GPU and simultaneously test the latency and power consumption. The power statistics, including GPU and DRAM power, are measured using the ‘nvidia-smi’ command in the terminal. Since the data storage and evaluation are performed with the SEAGATE (ST4000NM00A) HDD as storage media, the parameters from the SEAGATE datasheet [8] are used for calculation. The latency and energy consumption generated by the data access, writing/reading are estimated according to the interface access speed, max sustainable transfer rate, and average operating power, which can be acquired from the datasheet of 1.5 Gb/s, 215 MB/s, and 10.77 W, respectively. According to the above metrics, the latency and energy consumption for compressing and decompressing the VisualQA v2.0 dataset are calculated in Table R1.

Table R1. Latency and energy consumption of GPU-based processing system.

Performance	Process	GPU + DRAM	Storage
Latency (s)	Encode	115.97	124.8
	Decode	122.8	124.8
Energy (J)	Encode	1275.67	626
	Decode	1350.85	626

ASIC-based processing system:

The ASIC-based processing system consists of a hardware accelerator and off-chip DRAM and storage [9]. The recently reported CHIMERA accelerator (40 nm CMOS technology) [5] with 0.92 TOPS on peak performance and 2.2 TOPS/W on energy efficiency is adopted for the evaluation. DDR4 DRAM with 8 memory controllers and 64-bit channel per controller is utilized as external memory, which reported 45 pJ/bit on read/write energy and 65 ns/60 ns on read/write latency [10]. For data compression/retrieval, the latency/energy consumption can be obtained by dividing the number of operations by peak performance/energy efficiency. For the DRAM fetching process, the latency is calculated by multiplying the read/write latency per operation and the operation numbers, and the energy consumption can be achieved by multiplying the energy consumption per bit and the data size. For the VisualQA v2.0 dataset, the total number of operations during encoding/decoding is 900 G/400 G for the CAE network. The transferred data size after compression is 100 Gb. The parameters for HDD can be referenced in the GPU-based processing system section. Based on the above parameters, the evaluation results on latency and energy consumption can be estimated, as listed in Table R2. Although there are other ASIC and DRAM with relatively high performance, the improvement in latency and energy consumption of the compressed system based on ASIC is slight since the bottleneck is the data transfer and storage.

Table R2. Latency and energy consumption of ASIC-based processing system.

Performance	Process	ASIC	DRAM	Storage
Latency (s)	Encode	0.98	24.41	124.8
	Decode	0.43	24.41	124.8
Energy (J)	Encode	0.41	9	626
	Decode	0.18	9	626

Memristor-based near-storage in-memory processing system:

The architecture of the memristor-based near-storage in-memory processing system consists of 2133 storage banks and 8 in-memory computing banks for VisualQA v2.0 dataset compression and storage, as shown in Fig. R4. Each computing and storage bank include 16 cores. The size of the memristor array is set to be 1 Mb (1024×1024) [11] in the storage core, and that in the computing core remains 4 kb (256×16), as discussed in the manuscript. The latency and energy consumption are dominated by the memristor array, ADC, digital-to-analog converter (DAC), and buffer, mainly considered during evaluation [12-15]. During the compression process, the weights of the two-layer CAE network can be completely mapped into one computing core, so one bank can map 16 sets of weights and realize parallel computing. The dataset is compressed to 16.67×10^9 pixels with 6-bit accuracy for each pixel by the two-layer CAE network. In this case, the compressed data requires about 2133 storage banks for data storage, and different storage banks can perform parallel writing or reading of data. Accordingly, 8 computing banks are integrated during evaluation to boost speed.

Fig. R4 Diagram of memristor-based near-storage in-memory processing architecture. **a**, Computing bank and a memristor-based computing core. **b**, Storage bank and internal structure of a storage core.

All the circuit components in the near-storage in-memory processing system are evaluated under the 65 nm technology node. A 0.2 V read voltage with a 20 ns pulse width is adopted for the single-bit calculation of the memristor array. The mean weight value, extracted from the conductance distribution of all cells in the weight matrix, is used to calculate the energy consumption. Similarly, the conductance value used for data writing and reading is extracted from the conductance distribution of the compressed data matrix. For data writing, the pulse numbers used for multi-bit memristor programming are calculated according to the total programming pulses (Fig. R5). We use CACTI [16] to evaluate the latency and energy consumption of the buffer, which is designed to be 64 KB in size and 2 kb in bandwidth to allow for parallel computing of all 16 cores in the computing bank. The ADC in [4] is adopted, which reported an 8-bit resolution with a sampling rate of 450 MS/s. 16 source lines (SLs) operate in parallel with one ADC on each. We use the DAC data from [17], which

reported 136 mW for digital/analogue conversion under 20 GS/s. All the parameters are summarized in Table R3. According to these metrics, the latency and energy consumption for compressing and decompressing the VisualQA v2.0 dataset are calculated in Table R4.

Fig. R5 Statistical pulse numbers to program 4-bit memristor. Programming pulses of 100 cells are counted for each conductance state.

Table R3. Summary of metrics of key circuit modules in memristor-based computing bank used for evaluation.

Module	Energy	Latency
ADC	14.3 pJ/bit	2.2 ns/op
DAC	6.8 pJ/conversion	50 ps/op
Buffer	0.38 pJ/bit	1.04 ns/access
Array	9.2 fJ/cell (encoding) 8.2 fJ/cell (decoding) 17 fJ/cell (reading) 25 pJ/cell (writing)	20 ns/pulse @ reading 100 ns/pulse @ writing

Table R4. Evaluation results of latency and energy during data compression and retrieval with memristor-based near-storage in-memory processing system.

Performance	Process	Buffer	Array	ADC	Data write/access	Overall
Latency (s)	Encode	0.27	10.41	0.57	22.14	22.14
	Decode	0.3	11.71	0.64	0.67	11.71
Energy (J)	Encode	1.37	0.13	7.63	11.66	20.79
	Decode	0.46	0.04	8.58	0.7	9.78

The following revisions have been made in the manuscript.

- 1) Page 15-16, line 438 and line 447-450. The benchmark comparison results of GPU-based and ASIC-based processing systems are supplemented into Fig. 5d and 5e. The caption of Fig. 5 is modified accordingly.
- 2) Page 21-22, line 636-665. The evaluation methods and processes of GPU- and ASIC-based processing systems are added into “Methods-System overhead evaluation”.
- 3) Table R1 and Table R2 are replenished into supplementary information as Supplementary Table 4 and Table 5, respectively.
- 4) The schematic diagram in Supplementary Fig. 17 is modified accordingly.
- 5) Page 14, lines 400-420. Corresponding discussion on benchmark comparison results is added and modified in the manuscript, which is written as: “We compared the latency and energy consumption of the proposed system with that of other competitive technologies, including JPEG, PNG, autoencoder (AE), and CAE implemented in the server-grade CPU-based processing system, and CAE implemented in the GPU-based, ASIC-based, and non-near-storage in-memory processing systems (see Methods and Supplementary Fig. 17). Since traditional compression techniques usually contain various domain transformations and complex lossless coding schemes (such as Huffman coding), in most cases, the compression is better suited for execution on the CPU. As illustrated in Fig. 5d and 5e, the near-storage in-memory processing system effectively reduces the latency and energy consumption by more than 20× and 180×, respectively, compared with server-grade CPU-based processing systems, 5.6× and 91×, respectively, compared with GPU-based processing system, and 5.6× and 30×, respectively, compared with both ASIC-based and non-near-storage in-memory processing systems. The significant improvements in energy efficiency can be mainly attributed to the integration of memristor-based computing banks and storages banks within the same unit, which effectively reduces the latency and energy consumption caused by circuit module interfaces and internal transfer during data movement. It should be noted that the latency and energy reductions of ASIC-based processing systems is almost the same as those of non-near-storage in-memory processing system, which is due to the fact that in both systems, the overhead of data access during compression and decompression processes is much greater than that of data processing. The overall latency and energy consumption are dominated by the storage unit”.
- 6) The contents about benchmark comparison results in Abstract, Introduction, and Discussion are modified accordingly.
Page 1, lines 26-28. “by over 20×/5.6× and 180×/91×, respectively, compared with

the server-grade central processing unit (CPU)-based/the graphics processing unit (GPU)-based processing system”.

Page 3, lines 82-85. “5.6× and 91× compared with the graphics processing unit (GPU)-based processing system, and 5.6× and 30× compared with the application-specific integrated circuit (ASIC)-based and non-near-storage in-memory processing systems. In addition, the storage density could improve by more than 3 times”.

Page 16, lines 467-472. “The benchmark comparison results illustrate that the memristor-based near-storage in-memory processing system can achieve reductions in latency and energy consumption of over 20× and 180×, respectively, when compared with the server-grade CPU-based processing system. Similarly, it achieves reductions of 5.6× and 91×, respectively, when compared with the GPU-based processing system, and 5.6× and 30×, respectively, when compared with the ASIC-based and non-near-storage in-memory processing systems”.

Besides, the corresponding references, figure captions and descriptions in the manuscript and supplementary information are changed accordingly. These changes will not influence the content and structure of the manuscript. We did not list the changes here but marked them in the highlighted version.

Reference

- [1] Xiao, R. et al. An Energy Efficient Time-Multiplexing Computing-in-Memory Architecture for Edge Intelligence. *IEEE J. Explor. Solid-State Computat.* **8**, 111-118 (2022).
- [2] Liu, H. et al. A high linear voltage-to-time converter (VTC) with 1.2 V input range for time-domain analog-to-digital converters. *Microelectron. J.* **88**, 18-24 (2019).
- [3] Osheroff, P. et al. A highly linear 4GS/s uncalibrated voltage-to-time converter with wide input range. In *2016 IEEE International Symposium on Circuits and Systems (ISCAS)* (IEEE, 2016).
- [4] Tripathi, V. et al. An 8-bit 450-MS/s single-bit/cycle SAR ADC in 65-nm CMOS. in *2013 Proceedings of the ESSCIRC (ESSCIRC)* (IEEE, 2013).
- [5] Prabhu K. et al. CHIMERA: A 0.92-TOPS, 2.2-TOPS/W edge AI accelerator with 2-MByte on-chip foundry resistive RAM for efficient training and inference. *IEEE J. Solid-St. Circ.* **57**, 1013-1026 (2022).
- [6] Goyal, Y. et al. Making the v in vqa matter: Elevating the role of image understanding in visual question answering. in *Proceedings of the IEEE conference on computer vision and pattern recognition (CVPR)* (IEEE, 2017).
- [7] NVIDIA Developer. *CUDA Toolkit 11.3*, [Online]. Available:

<https://developer.nvidia.com/cuda-11.3.0-download-archive>.

- [8] SeaGate Technology. www.seagate.com/www-content/datasheets/pdfs/exos-7-e8-msft-data-sheet-DS1957-4M-1909US-en_GB.pdf.
- [9] Zhang, W. et al. Edge learning using a fully integrated neuro-inspired memristor chip. *Science* **381**, 1205-1211 (2023).
- [10] Aly M. M. S. et al. The N3XT approach to energy-efficient abundant-data computing. *Proc. IEEE* **107**, 19-48 (2018).
- [11] Lv, H. et al. BEOL based RRAM with one extra-mask for low cost, highly reliable embedded application in 28 nm node and beyond. in *2017 IEEE International Electron Devices Meeting (IEDM)* (IEEE, 2017).
- [12] Shafiee, A. et al. ISAAC: A convolutional neural network accelerator with in-situ analog arithmetic in crossbars. *ACM Comput. Archit. News* **44**, 14-26 (2016).
- [13] Liu, Q. et al. A fully integrated analog ReRAM based 78.4 TOPS/W compute-in-memory chip with fully parallel MAC computing. in *2020 IEEE International Solid-State Circuits Conference (ISSCC)* (IEEE 2020).
- [14] Dong, X. et al. A circuit-architecture co-optimization framework for exploring nonvolatile memory hierarchies. *ACM T. Archit. Code Op.* **10**, 1-22 (2013).
- [15] Zangeneh, M. et al. Design and optimization of nonvolatile multibit 1T1R resistive RAM. *IEEE Trans. Very Large Scale Integr. VLSI Syst.* **22**, 1815-1828 (2013).
- [16] Muralimanohar, N. et al. CACTI 6.0: A tool to model large caches. *HP laboratories* **27**, 28 (2009).
- [17] Kim, S. et al. A 65-nm CMOS 6-bit 20 GS/s time-interleaved DAC with full-binary sub-DACs. *IEEE T. Circuits-II* **65**, 1154-1158 (2018).

REVIEWERS' COMMENTS

Reviewer #3 (Remarks to the Author):

The authors have addressed my remaining questions and comments with additional technical backup and discussions. The quality of the manuscript has been improved as well. I support the publication of this work.

RESPONSE TO REVIEWERS' COMMENTS

Response to the Reviewer 3

Overall Comment:

The authors have addressed my remaining questions and comments with additional technical backup and discussions. The quality of the manuscript has been improved as well. I support the publication of this work.

Response:

We sincerely thank the Reviewer's support and the valuable time in reviewing our manuscript and providing constructive comments to help significantly improve the quality of our work. We are happy that the Reviewer is satisfied with our revision.